# Assembly and positioning of actomyosin rings by contractility and planar cell polarity

Ivonne M Sehring[1], Pierre Recho[2,3], Elsa Denker[1], Matthew Kourakis[4], Birthe Mathiesen[1], Edouard Hannezo[2,5]*, Bo Dong[6,7]*, Di Jiang[1]*

[1]Sars International Centre for Marine Molecular Biology, University of Bergen, Bergen, Norway; [2]Department of Physico-Chemistry of Living Matter, Institut Curie, Paris, France; [3]Mathematical Institute, University of Oxford, Oxford, United Kingdom; [4]Department of Molecular, Cellular and Developmental Biology, University of California, Santa Barbara, Santa Barbara, United States; [5]The Gurdon Institute, University of Cambridge, Cambridge, United Kingdom; [6]Ministry of Education Key Laboratory of Marine Genetics and Breeding, College of Marine Life Sciences, Ocean University of China, Qingdao, China; [7]Laboratory for Marine Biology and Biotechnology, Qingdao National Laboratory for Marine Science and Technology

**Abstract** The actomyosin cytoskeleton is a primary force-generating mechanism in morphogenesis, thus a robust spatial control of cytoskeletal positioning is essential. In this report, we demonstrate that actomyosin contractility and planar cell polarity (PCP) interact in post-mitotic *Ciona* notochord cells to self-assemble and reposition actomyosin rings, which play an essential role for cell elongation. Intriguingly, rings always form at the cells′ anterior edge before migrating towards the center as contractility increases, reflecting a novel dynamical property of the cortex. Our drug and genetic manipulations uncover a tug-of-war between contractility, which localizes cortical flows toward the equator and PCP, which tries to reposition them. We develop a simple model of the physical forces underlying this tug-of-war, which quantitatively reproduces our results. We thus propose a quantitative framework for dissecting the relative contribution of contractility and PCP to the self-assembly and repositioning of cytoskeletal structures, which should be applicable to other morphogenetic events.

*For correspondence:
eh508@cam.ac.uk (EH);
bodong@ouc.edu.cn (BD);
di.jiang@sars.uib.no (DJ)

**Competing interests:** The authors declare that no competing interests exist.

## Introduction

In many developmental and cellular contexts, actin filaments construct complex and highly dynamic structures to accomplish cell shape changes such as in migration and cytokinesis (*Munjal and Lecuit, 2014*). Correct positioning of the actin filaments is essential. In polarized migrating cells, actin flows posteriorly and becomes associated with myosin II at the trailing edge to propel the cell forward (*Cramer, 2010*). In cytokinesis of vertebrate cells, the equatorial ring is established in many incidences by a cortical flow of actin filaments (*Bray and White, 1988*) driven by myosin contractility and is concentrated at the equator to ensure correct cell division (*Cao and Wang, 1990*; *DeBiasio et al., 1996*; *Mayer et al., 2010*). Despite the importance of the proper positioning of the actin cytoskeleton, our understanding of how cell polarity contributes to the organization of the cytoskeleton, and vice versa, is still incomplete. In *Caenorhabditis elegans* early embryogenesis, a flow of cortical myosin and F-actin towards the anterior pole carries PAR polarity proteins, which in turn modulate the actomyosin dynamics (*Munro et al., 2004*; *Mayer et al., 2010*). Emerging evidence

**eLife digest** Animal cells can move, and cell movements are particularly important during the early stages of development, when the developing embryo rapidly changes shape. These movements depend on a network of fibers made up of a protein called actin. Just like an animal's skeleton, this network provides an internal scaffold for the cell and supports the cell's movements. Another protein called myosin works closely with actin and acts as a motor that drives these movements.

To study how cellular movements contribute to development, scientists often turn to simple, tube-like sea animals called sea squirts. These strange-looking creatures are distant relatives of humans and other animals with a spinal cord. All of these related creatures develop a long, rod-shaped structure called the notochord during the earliest stages of their development that is critical for forming the nervous system. Studying the development of the notochord is easier in sea squirts than other animals because the sea squirt's notochord is made up of just 40 cells arranged in a single file.

Now, Sehring et al. provide new details about the forces that shape the notochord cells in sea squirts. The experiments used microscopes and fluorescent markers to see what happens as the cells elongate to form the rod-like notochord, which stretches from the front to the back of the animal. This revealed that a ring made of actin and myosin forms near the front end of each cell and then migrates to the middle of the cell, stretching it along the way.

Sehring et al. then treated the cells with a drug that blocks myosin's motor-like ability. In these cells, the actin–myosin ring remains stuck in the front of the cell. Treating the cells at a later stage, that is, when the rings had already arrived at the center, led to the central rings moving back to the front end of the cell. Furthermore, when a mutant sea squirt that had cells without a distinct front or back was treated with the myosin-blocking drug, the ring ended up at random places in the cells. Together, these results suggest that the placement of the actin–myosin ring is determined by a tug-of-war between the pull of the front end of the cell and the myosin-driven force of the ring itself.

These findings reveal a general rule that can determine the position of cell's inner skeleton. Future studies will ask how the front end of the cell attracts and pulls the ring, and what this means for tissue growth and organ formation.

also point to a role for the Wnt/planar cell polarity (PCP) pathway in modulating cytoskeleton dynamics through its key mediators, Rho GTPases, which exert effects on actin polymerization and myosin contractility (*Schlessinger et al., 2009*), although the mechanisms underlying this cross-talk remain obscure. On the other hand, in vitro experiments on reconstituted cytoskeletal structures (*Surrey et al., 2001*), as well as recent mathematical models (*Kruse et al., 2005*; *Hannezo et al., 2015*) suggest that actomyosin gels could have the properties to self-assemble, but the applicability of these findings to in vivo situations is not yet clear. Therefore, the interplay between self-assembly and polarity signals that organize the cytoskeleton remains largely unexplored.

The *Ciona* notochord is a transient embryonic structure, which is composed of 40 post-mitotic cells that are arranged in a single file after convergent/extension (C/E). Following C/E, the coin-shaped cells undergo continuous elongation along the anterior–posterior axis (*Cloney, 1964*; *Miyamoto and Crowther, 1985*; *Jiang and Smith, 2007*; *Dong et al., 2009*), acquiring a drum shape (*Figure 1A*). Our previous studies show that an actomyosin contractile ring is present in the basal equator (*Dong et al., 2011*) and produces a circumferential constriction. The force generated by the constriction is transmitted three dimensionally from the basal cortex towards anterior and posterior lateral domains through an incompressible cytoplasm, driving notochord cell elongation (*Dong et al., 2011*; *Sehring et al., 2014*) (*Figure 1B,C*). The actomyosin ring is maintained by a bi-directional cortical flow and is under constant turnover in a manner remarkably similar to that of the cytokinetic ring during cell division. The position of contractile rings influences notochord cell shape and elongation. For example, in α-actinin mutants, the ring cannot maintain its position at the equator, and consequently, the cells fail to elongate but acquire an asymmetric shape (*Sehring et al., 2014*). However, the mechanism of positioning the contractile ring in the equator of the notochord cells is unknown. This question is also of crucial relevance to our understanding of cytokinesis, where the position of the actomyosin ring is critical for the cells to divide properly

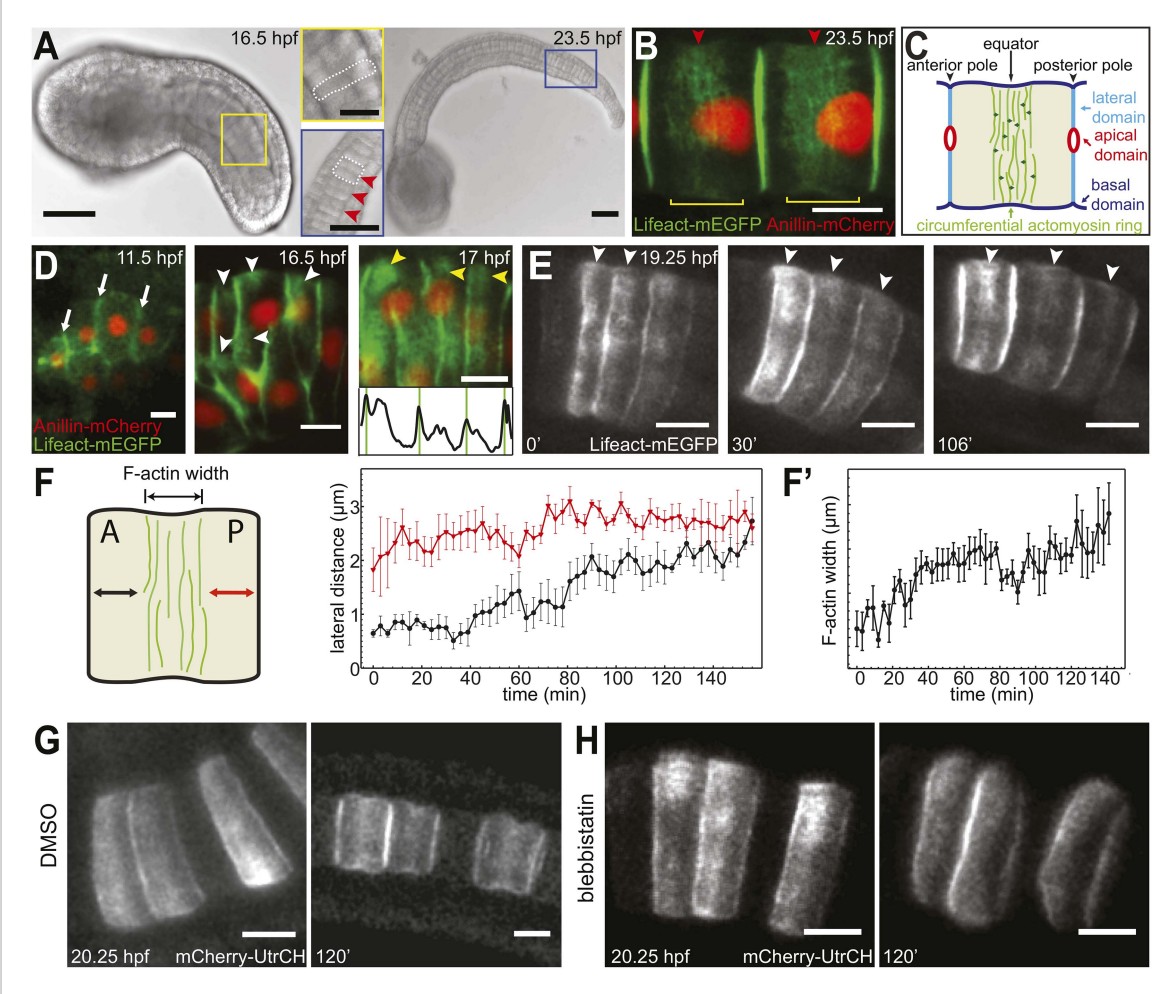

**Figure 1**. Establishment and relocation of anterior basal cortical actin filaments. (**A**) *Ciona* embryos at 16.5 and 23.5 hr post fertilization (hpf). Following cell intercalation, notochord cells at 16.5 hpf are coin-shaped (one is highlighted in the insert). At 23.5 hpf, cells are cylindrically elongated, and a circumferential constriction is present midway between the two poles (red arrowheads in insert). (**B**) Notochord cells are labeled with Lifeact-mEGFP (green) for actin and Anillin-mCherry (red) for the nucleus. Red arrowheads indicate the equatorial constrictions; yellow brackets outline the circumferential actin rings at the equatorial region. (**C**) A diagram of an elongating notochord cell at the onset of lumen formation with the nomenclature used in this paper. Small dark green arrows indicate the bi-directional cortical flow of actin filaments contributing to the construction of the actin ring. (**D**) Notochord cells labeled with Lifeact-mEGFP (green) for actin and Anillin-mCherry (red) for the nucleus. At the start of intercalation (11.5 hpf), actin is evenly distributed in the cell boundaries (white arrows). During cell intercalation, basal cortical actin patches (white arrowheads) appear adjacent to the anterior lateral domain. The actin patches begin to fuse next to the anterior pole of the cells (yellow arrowheads). The intensity was measured at positions of arrowheads. Vertical green bars indicate lateral domains. (**E**) Notochord cells expressing Lifeact-mEGFP for actin. These images are from *Video 1*. After cell intercalation, basal cortical actin patches (arrowheads) continue to fuse, forming a circumferential ring next to the anterior lateral domain, which subsequently relocates to the equator, as cells elongate. (**F**) Mean distances between the anterior lateral domain and the cortical actin ring (black), and the posterior lateral domain and the cortical actin ring (red) during cell elongation (*n* = 7; error bars = SEM). (**F'**) Mean ring width over time (*n* = 7; error bars = SEM). (**G, H**) Blebbistatin inhibits relocation of anterior basal cortical actin filaments and cell elongation. Notochord cells are labeled with mCherry-UtrCH for actin. Embryos are either treated with DMSO (**G**) or incubated in blebbistatin (**H**) for 120 min. Anterior to the left in all images. Scale bars in **A** represent 50 μm; in inserts, 20 μm; in **B–E**, **G**, **H** represent 10 μm.

The following figure supplement is available for figure 1:

**Figure supplement 1**. Establishment and relocation of anterior basal cortical myosin.

(*Sedzinski et al., 2011*) and to direct the distribution of cell-fate determinants correctly (*Clevers, 2005*; *Gómez-López et al., 2014*).

In addition, notochord cells acquire a subtle yet stable anterior/posterior (A/P) polarity: nuclei in all but the most posterior cell become localized at the posterior pole of the cell, while the classical PCP

protein *prickle* is localized at the anterior pole of the cell during the notochord cell elongation (*Jiang et al., 2005*; *Newman-Smith et al., 2015*). Whether the PCP pathway contributes to the process of cell elongation and whether and how PCP components affect contractile ring formation and positioning remains mysterious. In this study, we investigated the processes of actin ring formation and found, through dynamic imaging, physical modeling, as well as comparative and genetic analyses that the actomyosin contractility and PCP pathway work antagonistically to achieve a robust localization of the cytoskeleton.

## Results

### Posterior relocation of basal cortical actin rings to the equator in notochord cells

To analyze the development of the equatorial actin ring, we followed the expression of actin markers Lifeact-mEGFP and mCherry-UtrCH in notochord cells from the onset of C/E. Both Lifeact-mEGFP and mCherry-UtrCH bind to endogenous actin without interfering with its dynamics (*Burkel et al., 2007*; *Riedl et al., 2008*) and were shown previously to have the same localization pattern as the endogenous protein in *Ciona intestinalis* (*Sehring et al., 2014*). Before C/E (11.5 hpf, hours post fertilization), actin was uniformly localized in the basal cortex, with high concentration at notochord cell–cell contacts (arrows in *Figure 1D*). During cell intercalation (16.5 hpf), when the cells start to align into a column, the low and evenly distributed actin in the basal cortex was replaced by patches of cortical actin accumulations close to the anterior pole (white arrowheads in *Figure 1D*). With the alignment of the cells into a column (17 hpf), these actin patches connected into an actin ring spanning the entire circumference of the basal domain next to the anterior pole of the coin-shaped cell (yellow arrowheads and intensity graph in *Figure 1D*).

Time-lapse recordings revealed that this early anterior ring observed in coin-shaped cells is the precursor of the equatorial ring (*Video 1*). While the cells elongated, the ring relocated from the anterior pole to the equator of the cells (white arrowheads in *Figure 1E*). To ascertain if the relocation of the cortical actin ring was a continuous movement, we measured the distance between the edges of the ring and the two cell poles over time. While the length between the ring and the posterior pole grew only minimally, the length between the anterior pole and the ring increased steadily, until the distance from the ring to both poles was similar (*Figure 1F*). Concomitantly with ring relocation and cell elongation, the ring width increased (*Figure 1F'*). Myosin filaments (labeled with myosin regulatory light chain fused to mCherry, mCherry-MRLC) also formed a circumferential ring in the anterior basal cortex at the end of C/E. Similarly to the actin ring, the myosin ring subsequently relocated to the equator (*Figure 1—figure supplement 1*), where it remained for the rest of the elongation process.

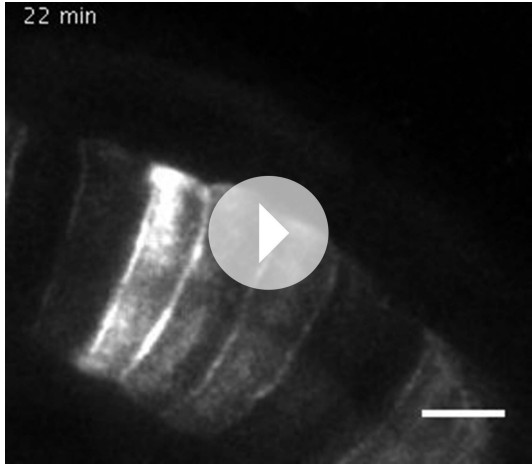

**Video 1.** Relocation of cortical actin during cell elongation. Notochord cells expressing Lifeact-mEGFP were recorded every min. Frame rate, 13fps.

### Actomyosin contractility is essential for equatorial positioning of the ring

Treatment with myosin II ATPase inhibitor blebbistatin at the onset of cell elongation prevented both cell elongation and the posterior relocation of the actin ring; instead, the actin ring remained at the anterior end of the cells (*Figure 1G,H*). This dependence of actin ring migration on myosin II activity prompted us to ask if long-term maintenance of the ring at the equator is also dependent on myosin II activity. We thus analyzed the effect of blebbistatin on cells that had already elongated substantially, and whose circumferential actin ring had been positioned at the equator (*Figure 2A*). At this stage, notochord cells in drug-treated embryos ceased to elongate (*Figure 2A,A'*). Surprisingly, the equatorial actin ring present at the start of the treatment (white arrowheads in *Figure 2A*) was lost; instead, we

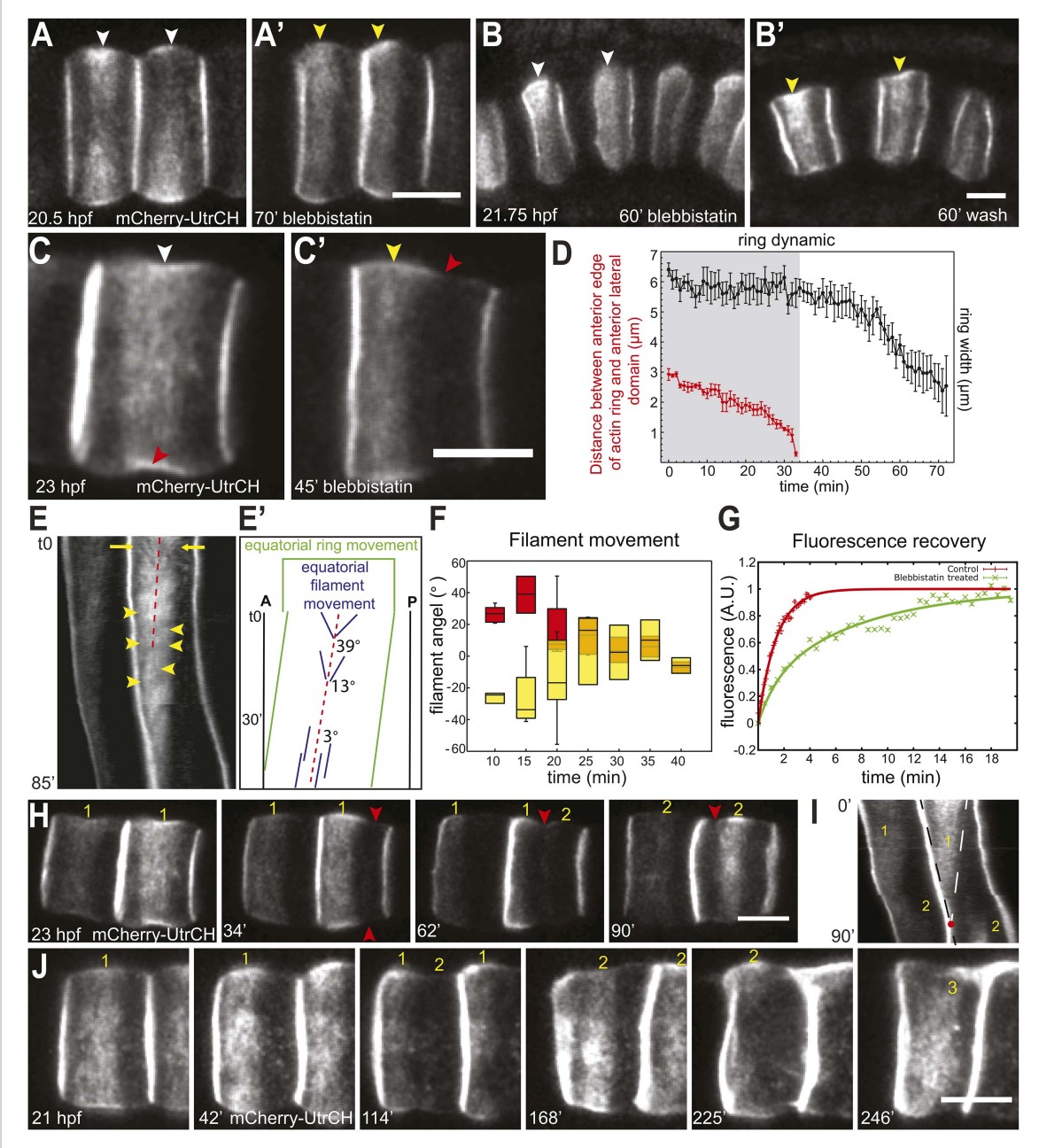

**Figure 2**. Shifting of equatorial actin filaments upon blebbistatin treatment. Notochord cells are labeled with mCherry-UtrCH or mCherry-hActin for actin. (**A**, **A'**) The equatorial actin ring (white arrowheads) in early elongating cells (**A**, 20.5 hpf) is relocated to the anterior pole (yellow arrowheads) after 70 min blebbistatin treatment (**A'**) (**B**, **B'**) The anterior relocation of the actin ring and inhibition of cell elongation after 60-min blebbistatin treatment (**B**) is reversed by a 60 min wash (**B'**). (**C**) At 23 hpf, the elongated notochord cell has a broad equatorial actin ring (white arrowhead) that is associated with a prominent constriction (red arrowhead). After 45-min blebbistatin treatment, the ring is shifted to the anterior pole (yellow arrowhead), whereas the constriction is not. (**D**) Mean distance between the anterior lateral domain and the middle of the ring (red), and the mean ring width (black) over time. While the ring shifts toward the anterior pole, indicated by the decrease of the anterior basal domain, its width stays relatively constant (shaded in gray). After the ring reaches the anterior lateral domain, the width decreases. n = 5; error bars = SEM. (**E**, **E'**) Kymograph of the shifting actin ring based on *Video 2*. Individual filaments (arrows) from the anterior and posterior edge move towards the center of the ring (the equator, indicated by the dashed line), which itself is shifting anteriorly. After a certain time, the equatorial bound movement of filaments becomes parallel to the movement of the ring (arrowheads). A diagram of the different movements is shown in **E'**. (**F**) Angles of single filament movement at specific times with respect to the center of the ring. Red (positive values), movement from posterior towards the center; yellow (negative values), movement from anterior towards the center. n = 2–7; error bars = SEM. (**G**) Fluorescence recovery after photobleaching in cells expressing mCherry-hActin. The entire actin ring region was bleached. Recovery is significantly slower in blebbistatin-treated cells. Control, n = 4; blebbistatin, n = 7; p = 0.012. Solid lines indicate a single exponential fit for the

*Figure 2. continued on next page*

*Figure 2. Continued*

control (red curve) with a turnover time of 90 ± 3 s, and a double exponential fit for the blebbistatin treatment (green curve), with a fast fraction with the same turnover as the control, and a slow fraction (f = 70% ± 4%) with a turnover time of 7.8 ± 0.6 min. (**H**) Time-lapse frames of *Video 3* showing the anterior movement of the equatorial ring (1), its disappearance, and the emergence of a second ring (2) at the equator. Red arrowheads indicate the circumferential constriction. (**I**) The kymograph illustrates the close succession of the second ring to the first ring. The red dot indicates the time when the first ring disappears at the anterior lateral domain, and the second ring begins to appear. (**J**) Time-lapse frames from *Video 4* showing the emergence of a second (2) and third (3) ring with long-term blebbistatin treatment. Anterior to the left in all panels. Scale bars, 10 μm.

The following figure supplements are available for figure 2:

**Figure supplement 1**. Effect of blebbistatin treatment.

**Figure supplement 2**. Talin localization at the equator is not affected by lower contractility.

---

observed an anterior accumulation of actin (yellow arrowheads in *Figure 2A′*), similar to the anterior concentration of actin in coin-shaped cells. The effect of blebbistatin is reversible (*Figure 2B,B′*): notochord cells were able to elongate significantly following a 60-min wash, and the cortical actin ring returned from the anterior edge of the cells to the equator.

To characterize the anterior shift of the equatorial actin ring triggered by blebbistatin treatment, we recorded the actin dynamics during drug treatment in elongated cells possessing an equatorial constriction and a broad actin ring. High-speed time-lapse recordings revealed a migration of the cortical actin ring from the equator to the anterior pole (*Figure 2C,C′*; *Video 2*), mirroring exactly the reverse sequence of migration of the ring towards the center in normal development. The front of the ring moved at a velocity of 91.8 ± 10.5 nm/min (n = 10). The speed of the shift was blebbistatin dose dependent; halving the blebbistatin concentration reduced the velocity significantly to 27 ± 0.5 nm/min (n = 10; p < 0.0001). The ring width stayed constant when it shifted anteriorly (*Figure 2D*). Remarkably, the encounter of the ring with the anterior lateral domain did not bring the movement to a halt. Instead, after the ring contacted the lateral domain (red line in *Figure 2D*), the posterior edge of the ring continued to move anteriorly at an increasing speed of 191.6 ± 21.4 nm/min (n = 10), so that the width of the ring narrowed (*Figure 2D*), until the entire ring disappeared at the position of the lateral domain (*Video 2*). The cell length did not significantly change during this process (*Figure 2—figure supplement 1A*; n = 10). We also examined the dynamic localization of talin, an actin-binding protein that bridges actin filaments and the adhesion apparatus at the cleavage furrow of dividing cells (*Sanger et al., 1994*; *Critchley, 2009*; *Kanchanawong et al., 2010*), and normally colocalized with the cortical actin ring at the equator in *Ciona* notochord (*Sehring et al., 2014*) (*Figure 2—figure supplement 2B*). Live imaging showed that talin concentrated at the cell equator slightly after the ring had been established centrally (data not shown), suggesting that talin actually responds to cortical repositioning rather than driving it. Upon blebbistatin treatment of already established central rings (*Figure 2—figure supplement 2A*), whereas the cortical actin was shifted to the anterior pole (*Figure 2—figure supplement 2A′*), talin lagged behind at the equatorial position (*Figure 2—figure supplement 2B′*). This result suggests not all components of the ring are shifted by blebbistatin.

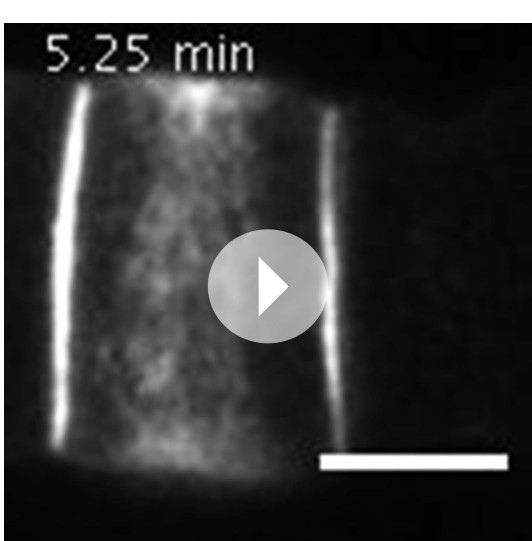

**Video 2.** Shifting of equatorial actin filaments upon blebbistatin treatment. Notochord cells expressing mCherry-UtrCH were recorded every 15 s. Frame rate, 13fps.

The actin ring in elongating notochord cells is highly dynamic and consists of circumferential filaments that flow to the equator from both sides of the ring (*Sehring et al., 2014*). The kymograph unveiled a persistence of this dynamics within the ring in blebbistatin-treated embryos: circumferentially oriented actin filament bundles (arrows in *Figure 2E*; further examples are shown in *Figure 2—figure supplement 1B*) from the anterior and posterior edges of the ring continued to flow towards the equator, which itself was shifting anteriorly (dashed line in *Figure 2E*). In the first 15 min of blebbistatin treatment, the filament bundles moved at a velocity of 7.62 ± 0.78 nm/s ($n = 18$), which was significantly slower than the velocity in control cells (DMSO-treated cells) (29.92 nm/s, $n = 10$; $p < 0.001$), where the filaments moved towards a stationary equator. Continued exposure to blebbistatin further reduced filament movement. Within the next 15 min, the velocity decreased significantly to 3.92 ± 0.9 nm/s ($n = 22$; $p = 0.004$). The deceleration of the filament bundles resulted in a flattening of their trajectories in the kymograph. Measurement of the angles of filament bundles revealed a reduction from 39° to 3° within 30 min of blebbistatin treatment (*Figure 2E',F*). After 31.1 ± 1.36 min ($n = 20$), directed movement of filament bundles towards the center of the ring (the moving equator) ceased, and only filament bundles moving nearly parallel to the direction of the shifting ring could be observed (yellow arrowheads, *Figure 2E*). After the ring had reached the anterior lateral domain, no pronounced filaments were detectable. There is no statistical difference in either velocities or angles between anterior- and posterior-directed filament bundles at any time. To analyze if there was still actin turnover within the ring at a time point when no prominent filament bundles were visible in a kymograph, or if it was a static ring shifting anteriorly, we performed fluorescence recovery after photobleaching (FRAP) experiments on notochord cells expressing mCherry-hActin. We bleached the entire equatorial ring in cells of control embryos, or the entire shifting ring in cells of embryos treated with blebbistatin for at least 30 min. For control embryos, the recovery curve was well fitted by a single exponential (*Figure 2G*), yielding a characteristic turnover rate of 90 s ± 3 s. In the shifting ring with blebbistatin treatment, fluorescence also recovered, indicating a persistent cortical flow of actin elements from outside the ring towards its center. Interestingly, the dynamics could no longer be fitted by a single exponential, as it contained two characteristic recovery times: a fast one, very similar to the control, and a much slower one of 7.8 ± 0.6 min (*Figure 2G*). A rough estimate of the flow-induced duration to close a bleached segment of 3 μm, assuming a mean velocity of 30 nm/s, is 100 s, which cannot be distinguished from the turnover time. However, after blebbistatin treatment, the mean velocity drops to roughly 6 nm/s (see Figure 6E), leading to a duration of 8 min, which is very similar to the measured slower recovery time. This further indicates a requirement of myosin motor activity for a strong cortical flow of actin elements, while local polymerization and depolymerization is still present and represents 30% of the recovery and appears to be very slightly affected by the blebbistatin treatment. These observations together reveal a persistence of fast inner-ring dynamics, albeit a bit slower because of decreased actin velocity, superimposed on a slow global shifting of the ring towards the anterior edge upon blebbistatin treatment.

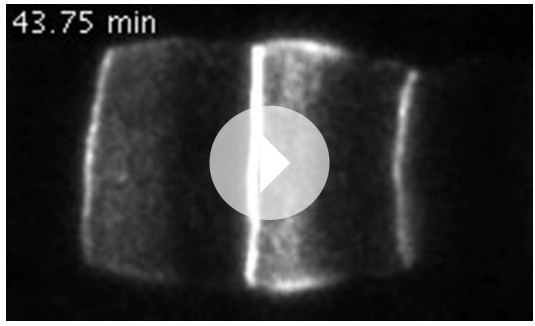

**Video 3.** Disappearance of the original actin ring and emergence of a new ring at the equator upon long-term blebbistatin treatment. Notochord cells expressing mCherry-UtrCH were recorded every 15 s. Frame rate, 13fps. DOI: 10.7554/eLife.09206.011

## Emergence of additional rings after prolonged blebbistatin treatment

Prolonged treatment with blebbistatin (>1 hr) led to the complete disappearance of the actin ring. Surprisingly, when the first ring was nearly gone, a new actin ring formed (*Figure 2H*, *Video 3*). A kymograph generated from the time-lapse reveals that the two rings overlapped in time only transiently: at the time point the first ring disappeared, the second ring emerged (*Figure 2I*). The position, shape, and size of the second rings were often less precise and less sharp than the first rings (*Figure 2H,J*). While the first ring disappeared after 106.67 ± 6.06 min of blebbistatin treatment ($n = 6$), the second ring had a significantly shorter dwelling time of only

61.33 ± 7.81 min (p = 0.001) before it disappeared at the anterior edge of the cells. Intriguingly, we consistently observed the dislodging of the actin rings from the morphological constriction (*Figure 2H*, arrowheads; *Figure 2—figure supplement 1C*), instead, the moving actin rings were associated with a morphological bulge at the basal domain (yellow arrowhead in *Figure 2C′*), and cells formed a new circumferential constriction (red arrowheads in *Figure 2C′,H*) in the wake of the shifting ring.

In cells that survived prolonged treatment of blebbistatin with relatively normal morphology, we observed the emergence of a third ring at the equator, after the disappearance of the second ring at the anterior lateral domain (*Figure 2J*; *Video 4*). In *Video 4*, the second and third ring appeared at 151 and 238 min of treatment, respectively.

## Anterior migration of actin rings depends on PCP in notochord cells

As notochord cells are planarly polarized, and ring migration is always unidirectional in wild-type embryo, this prompted us to examine the effect of A/P polarity on actin ring migration. A conspicuous feature of the A/P polarity is the posterior localization of the nucleus, which is regulated by *prickle*, a core PCP component. The *Ciona savignyi* mutant *aimless* carries a deletion in *prickle*, resulting in a loss of A/P polarity manifested in the randomized localization of nucleus in addition to an earlier convergent extension defect (*Jiang et al., 2005*). We first explored the possibility that the posterior nucleus might influence the direction of ring movement mobilized by blebbistatin. To this end, we examined the notochord of the ascidian *Halocynthia roretzi*, which follows a remarkably similar early developmental process as in *Ciona* (*Figure 3A,B*), except at the cell elongation stage, the nuclei are positioned in the center of the cells (asterisks in *Figure 3B*). A conspicuous circumferential constriction is present at the equator, which is colocalized with a cortical ring of actin (*Figure 3C*) and activated myosin (*Figure 3D*), whose activity is essential for notochord cell elongation (*Figure 3E,F*). We next treated embryos with elongating notochord cells with blebbistatin for 3 hr. Similar to what was observed in *Ciona*, the actin ring was shifted invariably anteriorly (*Figure 3G*), indicating that the position of the nucleus does not influence the dynamic behavior of the ring. We thus used *aimless* embryos to explore the role of a compromised A/P polarity on the repositioning of the ring and its shifting upon blebbistatin treatment, independent from its influence on nuclear position. In wild-type *C. savignyi* embryos, elongated notochord cells possess a circumferential actin ring at the cell equator and a posterior nucleus (*Figure 4A*). Blebbistatin treatment shifts the ring towards the anterior pole (*Figure 4B*), mirroring exactly the events in *C. intestinalis*. In *aimless* embryos, the intercalation of notochord cells (outlined in *Figure 4C*) is impaired, except in the posterior region where the cells often align into a single file (*Jiang et al., 2005*). In these cells, the loss of A/P polarity is evident by the random position of the nuclei (*Figure 4C* insert). In 92% of mock-treated *aimless* cells in this region, the actin ring is positioned at the equator (*Figure 4C* insert), showing that equatorial ring positioning is independent from PCP. However, reducing contractility in polarity-deficient mutant cells led to a dramatically different phenotype: after 60-min blebbistatin treatment, no unidirectional migration towards the anterior was observed. The direction of the shift was randomized and independent of the localization of the nucleus: examples for anterior nucleus and ring, anterior nucleus and posterior ring, posterior nucleus and ring, and posterior nucleus and anterior ring could be found (*Figure 4D,D′*). In 56% of the cells, the ring was still positioned at the equator, and in the rest, the ring migrated either to the anterior or posterior side (18% and 26%) (*n* = 57) (*Figure 4E*). These results imply that A/P polarity by *prickle* is not necessary for the

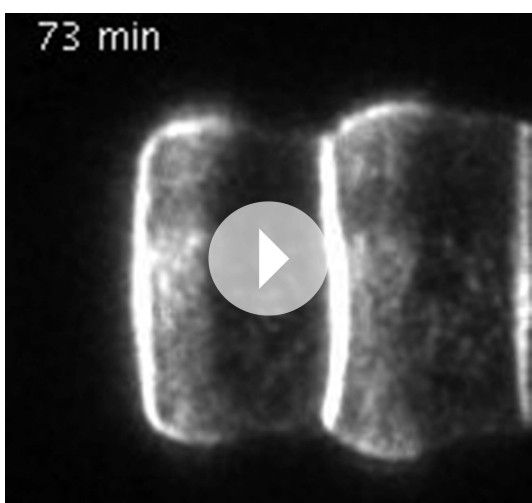

**Video 4.** Emergence of a second and third actin ring at the equator upon long-term blebbistatin treatment. Notochord cells expressing mCherry-UtrCH were recorded every 3 min. Frame rate, 13fps.

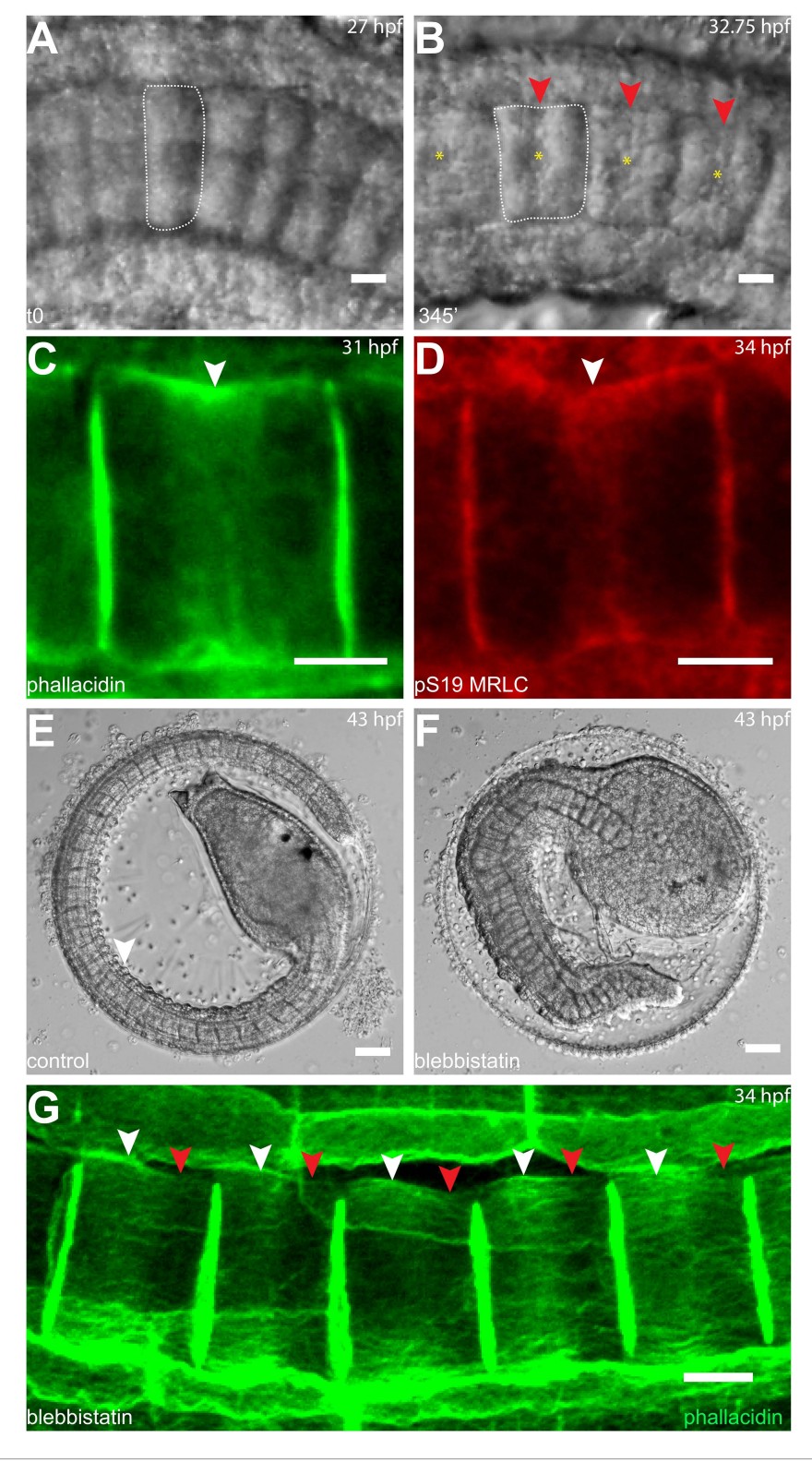

**Figure 3**. Circumferential actin rings are shifted anteriorly in *Halocynthia roretzi* notochord cells with centrally localized nuclei. (**A**, **B**) *Halocynthia* notochord cells elongate from coin-shaped (**A**) to drum-shaped (**B**).
A circumferential constriction appears at the equator of the cylindrical cell (arrowhead). The nucleus (asterisk) is localized in the center of each cell. (**C**, **D**) Cortical F-actin (arrow in **C**) and MRLC (arrow in **D**) accumulate at the
*Figure 3. continued on next page*

*Figure 3. Continued*

equatorial region of the basal domain. (**E**, **F**) Notochord elongation (**E**, DMSO-treated) is abolished in *Halocynthia* embryos treated with 100 μM blebbistatin for 16 hr (**F**) starting at the onset of cell elongation (27 hpf). (**G**) Circumferential actin rings (white arrowheads) are shifted anteriorly after 3-hr blebbistatin treatment at 31 hpf (13°C). Similar to what is observed in *Ciona* notochord cells (*Figure 2C', H*), the shifted ring is associated with a circumferential bulge (white arrowheads), whereas the constriction is located posterior to the ring (red arrowheads). These results indicate (1) a conservation of the equatorial actomyosin contractile mechanism to drive notochord cell elongation in *Halocynthia*, (2) that the position of the ring does not influence the position of the nucleus, (3) the position of the nucleus does not influence the position of the ring, and (4) the nucleus position does not affect the direction of the ring shift. Anterior to the left in panels **A–B**, **G**. Scale bars in **E** and **F**, 50 μm; in all others, 10 μm.

establishment of an equatorial actin ring; however, it is instrumental for the direction of movement of the ring upon blebbistatin treatment.

## A simple biophysical model of the actomyosin cortex for the formation and maintenance of equatorial actomyosin rings

In order to probe quantitatively these findings, we follow the theory of active gels (*Kruse et al., 2005*; *Prost et al., 2015*) to develop a very simple biophysical model of the actomyosin cortex as a viscous contractile gel, undergoing steady turnover (see Appendix 1: Physical modeling of the *Ciona* cortical flows for details). Such models generically predict spontaneous accumulations of actomyosin. Indeed, if the actomyosin concentration is slightly higher in a given region, then the contractile stress is also locally higher compared to the surroundings. Because of this initially small imbalance, surrounding actomyosin fibers flow towards the accumulation, making it even denser and even more contractile (*Recho et al., 2013*). This self-reinforcing loop, which concentrates actomyosin in a single spot at the cortex with filaments flowing towards the ring, is resisted by their depolymerization and effective diffusion (*Figure 5A*), which favor a homogenous cortex.

The cell cortical layer of actomyosin is modeled as a thin axisymmetric layer of length $L$ with principal axis $z = -L/2$ (anterior side)…$L/2$ (posterior side). Because the thickness of the cortex is two orders of magnitude below cell size, we can use a thin film lubrication approximation and do not resolve the cortex radial direction. Therefore, all equations are invariant in the radial and orthoradial direction and the model is one dimensional. The simplified chemo-mechanical problem then consists of three equations equipped with appropriate boundary conditions:

• Conservation of the actomyosin density ρ through a classical reaction-drift-diffusion process reads:

$$\partial_t \rho + \partial_z (\rho \nu) - D\partial_{zz}\rho = \frac{\rho_0 - \rho}{\tau}, \tag{1}$$

where $\nu$ denotes the actomyosin velocity. We have denoted $\tau$ the turnover time, $D$ an effective diffusion coefficient of actomyosin accounting for non local turnover and $\rho_0$ the target density.

Boundary conditions prescribe the actin fluxes $J = \rho\nu - D\partial_z\rho$ at $z = -L/2$ and $L/2$.

Force balance neglecting friction with the extracellular matrix reads

$$\partial_z \sigma = 0, \tag{2}$$

where σ is the mechanical stress in the actomyosin meshwork. The neighboring cells impose a given residual mechanical stress $\bar{\sigma}$ on the cell boundaries. In the absence of friction, as indicated by *Equation (2)*, the mechanical stress is homogeneous and $\sigma(z) = \bar{\sigma}$.

• At last, we prescribe the constitutive behavior of the gel as

$$\sigma = \eta\partial_z \nu + \chi\rho. \tag{3}$$

We have denoted $\chi$ the contractility arising from myosin motors, and $\eta$ the viscosity of the gel. We set $\bar{\sigma} = \chi\rho_0$, such that the value of the residual stress at low contractility, in the absence of boundary fluxes $(J(\pm\frac{L}{2}) = 0)$, ensures a homogenous distribution of actin as observed experimentally before elongation (16.5 hpf, *Figure 1A*). Indeed, $\rho = \rho_0$, $\nu = 0$, and $\sigma = \bar{\sigma}$ is then a trivial solution of *Equations (1–3)*.

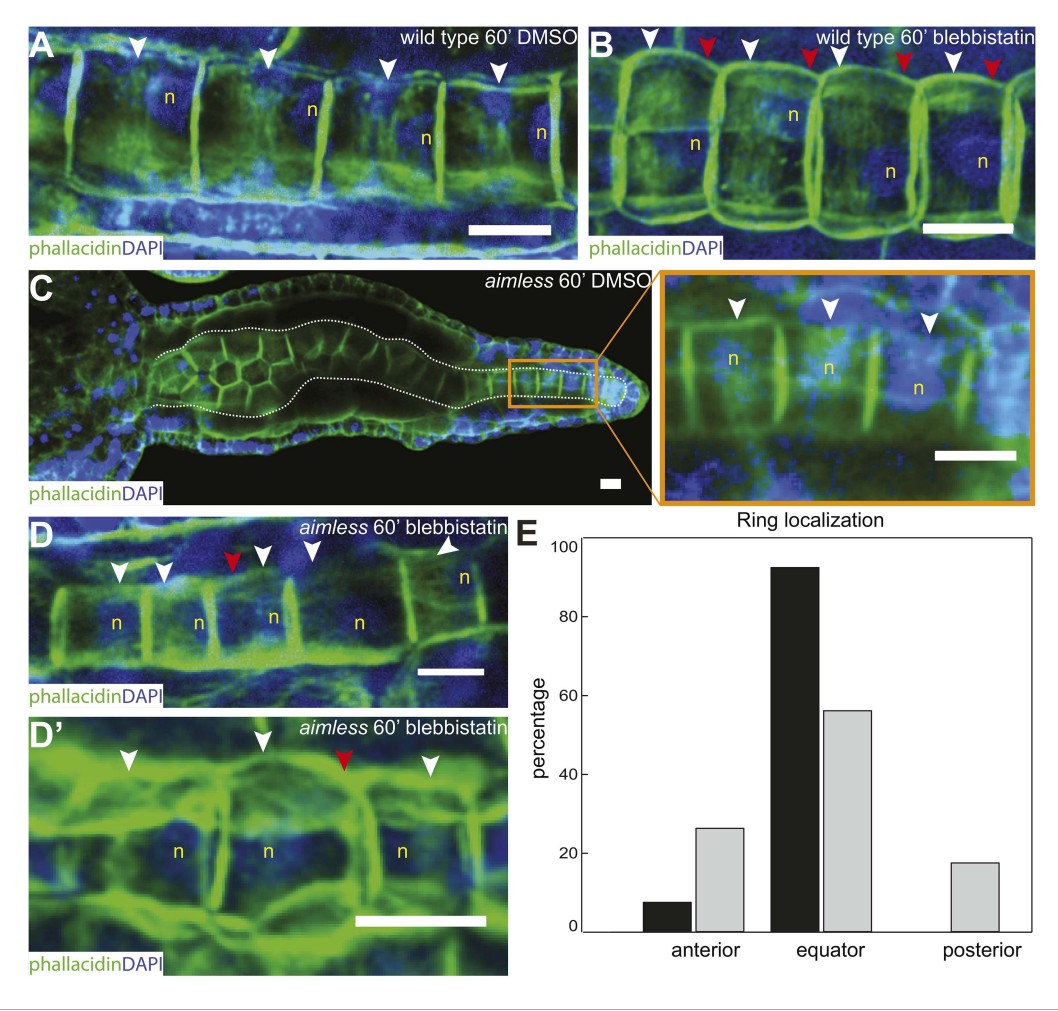

**Figure 4**. Anterior shifting of the actin ring is disrupted in the *prickle* mutant *aimless*. (**A–D'**) *Ciona savignyi* embryos are stained with phallacidin for actin and DAPI for nuclei. The actin ring (white arrowheads) is positioned at the equator in control notochord cells (**A**) and is shifted anteriorly by 60 min blebbistatin treatment (**B**), whereas the posterior localization of nucleus (n) is not affected by blebbistatin. Red arrowhead indicates the constriction. (**C**) Notochord in an *aimless* embryo (outlined by dashed line) with impaired cell intercalation in the anterior region, and fully intercalated and significantly elongated cells in the posterior region. The actin ring in these cells is localized at the equator, but the nucleus is placed in a random position (insert). (**D**, **D'**) 60-min blebbistatin treatment mislocalizes the actin ring in *aimless* embryos. (**E**) Distribution of actin rings in mock-treated (black; *n* = 53) and blebbistatin-treated (gray; *n* = 57) *aimless* cells. Anterior to the left in all panels. Scale bars, 10 μm.

For the problem to be fully specified, we still need to impose the values of the boundary fluxes of actin $J(\pm\frac{L}{2})$. To begin, in order to expose the role of contractility only, we first assumed that they vanish. It should be noted that the actomyosin flux $J$ encompasses both actin filament velocity, and an effective diffusive flux arising from actin polymerization (see Appendix 1: the model). Therefore, a vanishing total flux does not necessarily entail a vanishing velocity. Then, a linear stability analysis of *Equations (1–3)* predicts a threshold of actomyosin contractility $(\chi\rho_0)_c = \eta\left(\frac{1}{\tau} + \frac{4\pi^2 D}{L^2}\right)$ above which, the homogeneous cortex loses stability and a mechanically stable central ring forms even in the absence of external signaling cues (see Appendix 3: steady states, and *Figure 1* in Appendix 1: the model for the stability diagram and details on the boundary conditions). We can then interpret that the driving force positioning the ring at the equator is the contractility increase during the process of ring migration. Before finalizing the model, we check two of its key assumptions that (1) contractility increases smoothly as ring migration proceeds, and (2) the velocity gradient of actin filaments towards

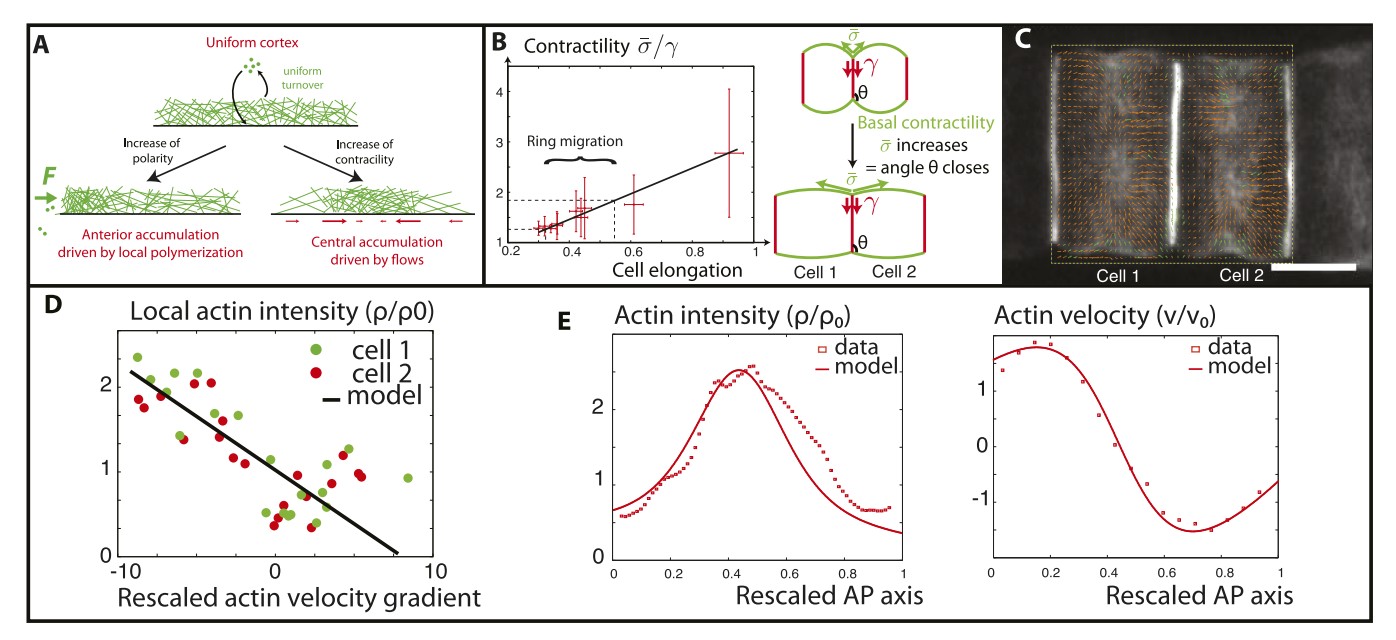

**Figure 5**. Verifications of the model assumptions and fitting of parameters. (**A**) Sketch of our model. Contractility destabilizes an initially homogenous cortex into a central ring, whereas PCP-driven preferential anterior polymerization localizes the ring on the edge. (**B**) Measurement of the angle between lateral and basal membrane during the elongation (2.5 increase) and ring migration (1.5 increase) process, which indicates their relative tensions (n > 15 for each time point). Basal tension increases with time. $\bar{\sigma}$ is the basal contractility, $\gamma$ is the lateral contractility and $\theta$ is the angle between lateral and basal membranes. We have the geometric relation $\bar{\sigma} \cos\theta = \gamma$. (**C**) PIV analysis of cortical flows in late stage embryos. (**D**) Linear negative correlation between local actin intensity and velocity gradients, as extracted from PIV. Actin intensity and velocities have been rescaled in the dimensionless units described in the main text and in Appendix 2: rescaling. (**E**) Comparison between intensity and velocity profiles and our theoretical predictions (data extracted from **C**). The velocity field is rescaled by the average velocity.

the center should depend linearly on the local contractility, indicated by local actomyosin concentration (as seen in *Equation 3*). To address the first assumption, we measure the angle between the lateral and basal side, from embryos at various stages (n > 20 angles for each embryo) (*Figure 5B*). As shown recently (*Maître et al., 2012*), this angle $\theta = \arccos\left(\frac{\bar{\sigma}}{\gamma}\right)$ reflects a force balance between the tensions of the basal ($\bar{\sigma}$) and lateral ($\gamma$) surfaces, and therefore can be used as a proxy for tension changes. Interestingly, the angle decreased smoothly and continuously during elongation and ring migration, indicative of an increased basal tension relative to lateral tension (*Figure 5B*), by roughly a factor of 2.5 during the elongation process, and 1.5 during the ring migration process. We assume at first order approximation that lateral tension is constant and set in the model that ($\bar{\sigma}$) increases by a factor 1.5. As we shall show later, such an increase enables a good fit to all of the available data (see Appendix 4: rough estimates of model parameters, and Appendix 5: model predictions for further details). To test the second assumption, we performed PIV analysis on high frequency movies of actin flows (*Figure 5C*), to measure local velocity gradients. When plotting these as a function of local actin concentration, we found a robust negative linear correlation (*Figure 5D*), which shows that flows are driven by differences in actomyosin concentration, validating quantitatively *Equation 3* of our model. From the slope of the correlation, as well as from the characteristic of actin bundles measured above in the kymographs, we could extract the ratio ($\chi/\eta$) of contractility and viscosity. Finally, we fixed the last parameters of our model through FRAP experiments ($\tau$ = 90 s) and through the width of the ring intensity profile (D = 2.10$^{-12}$ m²/s) (see Appendix 4: rough estimates of model parameters for the details of parameter fitting). Under these assumptions and with these parameters, a central ring spontaneously forms at the center of the cell when contractility increases above the critical threshold $(\chi\rho_0)_c$, and contractility is self-sufficient to maintain the ring structure through actomyosin flows. However, the experimental data show the existence of a stage where the ring is positioned at the anterior side and also underlines the importance of PCP in the repositioning

of actin rings. Our final model therefore incorporates polarity in the model in the simplest way possible: by assuming that it creates a small preferential polymerization of actin at the anterior side, that is, there is a non-zero flux at the anterior side, different from the flux at the posterior side. We considered that the flux on the posterior side was still zero, that is, $J(-L/2) = F$ and $J(L/2) = 0$. We showed in *Figure 7* in Appendix 5: model prediction that lifting this constraint does not qualitatively change our results, as the key parameter is the difference between anterior and posterior flux, but not their respective magnitude and our system is locally robust with respect to this type of perturbation. Assuming the existence of a preferential polymerization at one boundary due to PCP was supported by the well-studied link between PCP and actin polymerization (*Wallingford and Habas, 2005*). In particular, Disheveled (Dsh), one of the core member of the PCP pathway has been shown to activate key actin regulators such as Rho and Rac (*Tahinci and Symes, 2003*; *Wallingford and Habas, 2005*), as well as Daam1, a member of the formin protein family (*Kida et al., 2007*; *Gao and Chen, 2010*). It should be noted that as we are treating the actomyosin gel as a single species (with the assumption that bipolar filaments performing the contractile power stroke co-localize with actin), assuming that PCP localizes myosin anteriorly, as reported in *Newman-Smith et al., 2015*, would yield the same qualitative results. With these final boundary conditions, the dynamical system *Equations (1–3)* predicts a transition (which is now smooth, see *Figure 1* in Appendix 1: the model) between two mechanically stable states of the actin ring: a central position if the contractility $\chi$ is large enough, and an anterior position when $\chi$ is small enough and the polarity-induced actin flux F dominates. We set the value of F using the filament velocity order of magnitude as well as the experimental actin density profiles when the contractility is impaired (blebbistatin experiments, see Appendix 5: model predictions).

To verify the model, we then numerically integrated *Equations (1–3)* in order to calculate the steady state of a central actin ring, using the parameters deduced above. We found that our model can reproduce quantitatively very well both the velocity and intensity profiles in the entire actin cortex at the late stages of elongation (*Figure 5E*). Interestingly, we predicted, and observed in the PIV, a rather large, non-zero value for the actin filament velocity at the posterior edge of the cell, which is compensated in the model by actin polymerization on the side, since the total flux is zero. Moreover, we note a remaining, albeit small bias in ring position towards the anterior side, as in the data, showing that an anterior-imbalance remains at the later stages, both in simulations and in the data.

Next, we wished to compare the dynamics of ring migration predicted by the model to the experimental data, both in the case of normal ring migration towards the center, and blebbistatin-induced ring migration towards the anterior side. We simulated the effect of the linear increase in contractility, deduced from *Figure 5B*, and followed the appearance and migration towards the center of a ring. After 110 min, we then simulated a blebbistatin treatment as an exponential decrease in contractility, following previous experimental results on traction force decrease with blebbistatin (*Lam et al., 2012*; details in Appendix 5: model predictions), which predicted a rapid migration back towards the anterior pole. *Figure 6A* displays a kymograph of the numerical integration. Next, we compared model predictions and experimental data (average over 7 cells for normal ring migration, and 5 cells for blebbistatin treatment). It should be noted that in our model, the migration speed of the ring is controlled by the dynamics of contractility changes. This is in agreement with our experimental observation that the velocity is dependent on the dose of blebbistatin used, but also with the fact that the velocity of the ring migration is an order of magnitude smaller than the velocity of the actin filament bundles.

Moreover, a simple order of magnitude calculation shows that this decrease in the velocity of actomyosin explains the FRAP recovery curves (*Figure 2G*). Indeed, in control cells, a rough estimate of the time needed to close a bleached segment of 3 µm through flows, given a mean velocity of 30 nm/s, is 100 s, which cannot be distinguished from the turnover time. Therefore, we only see a single-exponential recovery. However, after blebbistatin treatment the mean velocity drops to roughly 6 nm/s (see *Figure 6E*), leading to a time of 8 min for flow-induced recovery, which is strikingly similar to the time extracted from the FRAP analysis of 7.8 min. There are then two time scales for recovery, one linked to turnover, unaffected by blebbistatin, and one linked to flows, which is dramatically compromised by blebbistatin.

We first examined the dynamics of normal ring migration to the center and verified that our dynamical prediction for the positions of the anterior and posterior sides of the ring matched very well the experimental data (*Figure 6B*). We also compared theoretical and experimental profiles of actin

intensity during ring migration towards the center (*Figure 6C*), and observed again a good agreement between the two: first an exponential ring profile at the anterior pole, then detachment of a ring of broad thickness during migration, and then refinement of a thinner central actin ring.

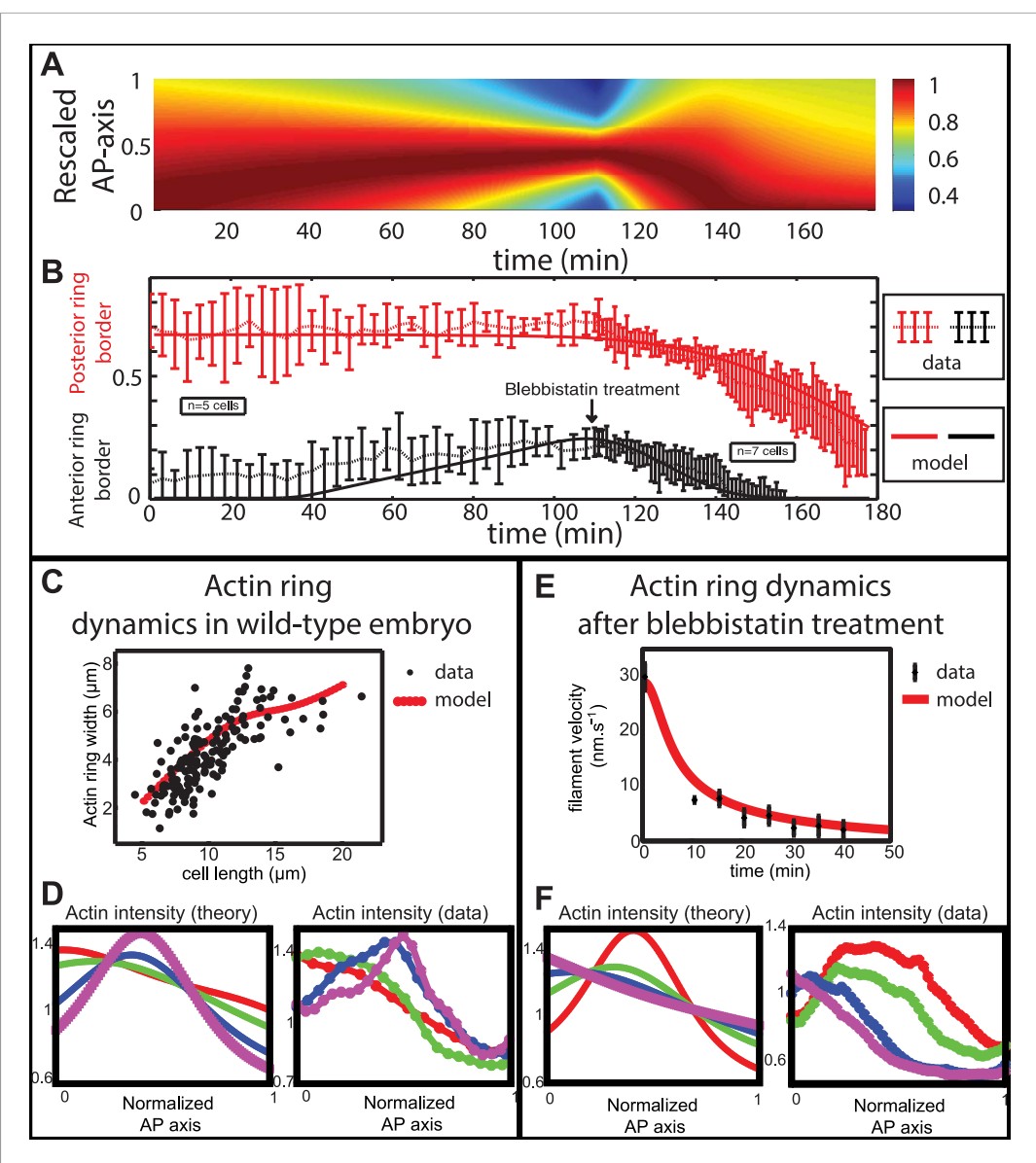

Figure 6. Comparison between theory and experiments on the dynamics of actin rings during migration and blebbistatin treatment. All theoretical curves are extracted from the same parameter set (see Appendix 5: model predictions for details on non dimensional values and parameters). (**A**) Kymograph of actin intensity during central ring migration (left part) and during blebbistatin treatment starting at 110 min. The color code indicates local actin intensity. (**B**) Comparison of the model and experimental anterior and posterior lateral domains during normal development (data taken from the average of 5 cells) and blebbistatin treatment at 110 min (data taken from the average of 7 cells). The y-axis indicates the position of the anterior and posterior border (defined as 50% of the ring maximal intensity). (**C**) Actin ring width vs with cell length, throughout cell elongation. The thick line is our theoretical prediction. The black dots are the measured data ($n = 7$; error bars = SEM). (**D**) Theory-to-experiment comparison of actin intensity profiles during central ring migration: 0 min (red), 30 min (green), 60 min (blue), 80 min (purple). (**E**) Theory-to-experiment comparison of filament velocity following blebbistatin treatment. (**F**) Theory-to-experiment comparison of actin intensity profiles after blebbistatin treatment: 0 min (red), 15 min (green), 30 min (blue), 45 min (purple).

Moreover, a rather intriguing feature of our model is that it predicts the size of the actin ring should first increase linearly with the length of the cell, before showing a plateau region for large cell sizes (*Figure 6C*). This is because ring formation is a spontaneous, self-organizing phenomenon that depends on boundary conditions, and therefore on the size of the cell. We measured experimentally actin ring width during cell elongation, and found it indeed increases linearly with the length of the cell, in very good quantitative agreement with our model, with all parameters having already been fixed above. In cytokinesis, it was proposed that such a scaling could allow cytokinetic time to be independent of cell size as observed in *C. elegans* embryos (*Turlier et al., 2014*). Our model provides a natural explanation to how this might be implemented in a simple manner in vivo.

Next, we turned to the blebbistatin treatment. The only free parameter is the timescale of contractility decrease (over a time of 15 min), which we fitted through ring position shifting, keeping all other parameters constant. This yielded a good theory-experiment agreement for the shifts of both the anterior and posterior sides of the ring (*Figure 6B*). Having fixed the one parameter for the blebbistatin treatment, we sought to test our model further via second independent measurement. We plotted the predicted decay of actin velocity from our simulations and compared it to the experimental data measured above. Again, we found an excellent quantitative agreement between predictions and data (*Figure 6E*). Finally, we examined the spatial profiles of actin intensity during blebbistatin treatment and migration towards the anterior edge. This recapitulated in reverse the normal migration sequence and showed good qualitative agreement between theory and experiments.

To verify the role of PCP in the model, we performed the simulation under the conditions of no polarity and random polarity (*Figure 7A*): for low polarity and increased contractility (*aimless* mutant), the ring formed directly in the center of the cell; for low contractility and low, uncoordinated polarity between neighboring cells (*aimless* mutant with blebbistatin), the actin ring did not have any strong forces positioning it, and was therefore positioned randomly, depending on the strength of the respective anterior and posterior boundary fluxes (see Appendix 5: model predictions for further details).

We next turned back to our data to find experimental clues for this preferential polymerization of actin at the anterior side. As noted earlier, Dsh is a core member of the PCP pathway (*Keys et al., 2002*), and we found that it initially accumulated at the basal membrane, but was relocated to the lateral domains, with a preference for the anterior side (28% ± 9% enrichment compared to the posterior side, n = 11 cells) (*Figure 7B*), mirroring the localization of Prickle protein at this stage (*Figure 7C*) (*Jiang, et al., 2005*; *Newman-Smith, et al., 2015*). Interestingly, this relocation occurred roughly at the same time as the appearance of the anterior ring (between 19 and 20 hpf). This is fully in line with our prediction of a PCP-driven preferential polymerization at the anterior side, which could be mediated by Dsh.

## Discussion

Ascidians have the simplest notochord in the Chordate phylum. It consists of merely 40 cells arranged in a single file at a transient stage of development. Yet, emerging evidence suggests that the *Ciona* notochord possesses an intriguing array of complexity, including differential gene expression, asymmetrical cell lineages, desynchronized morphogenetic behavior (*Reeves et al., 2014*), and an A/P polarity which manifests in the posterior localization of nuclei in the first 39 cells (*Jiang et al., 2005*). The current study reveals hitherto unknown aspects of A/P polarity in dynamical properties of actomyosin cortex. An actin ring forms at the anterior side of cells as core PCP proteins such as Disheveled relocate there, and then migrates to the equator as contractility increases, where it contributes to cell elongation. Interestingly, planar polarized contractile flows were reported during *Drosophila* germ-band extension (*Rauzi et al., 2010*). The ring movements that we observe are very robust, with surprisingly little fluctuation, given the highly dynamic nature of the ring. Here, we could disentangle the contributions of contractility and PCP to this robust positioning, by a combination of drug and genetic experiments. We show that decreasing contractility via blebbistatin reverses the normal developmental sequence, causing an anterior shift of the equatorial ring (*Figure 2*), which can be observed consistently throughout the elongation stage, but not in polarity mutants. This suggests a tug-of-war between contractility and cellular polarity: sufficiently large myosin contractility antagonizes the *Prickle/Dishevelled*-mediated anterior bias and can self-organize the actin ring in the center of the cell purely by physical forces, as shown by our theoretical analysis (*Figure 7A*).

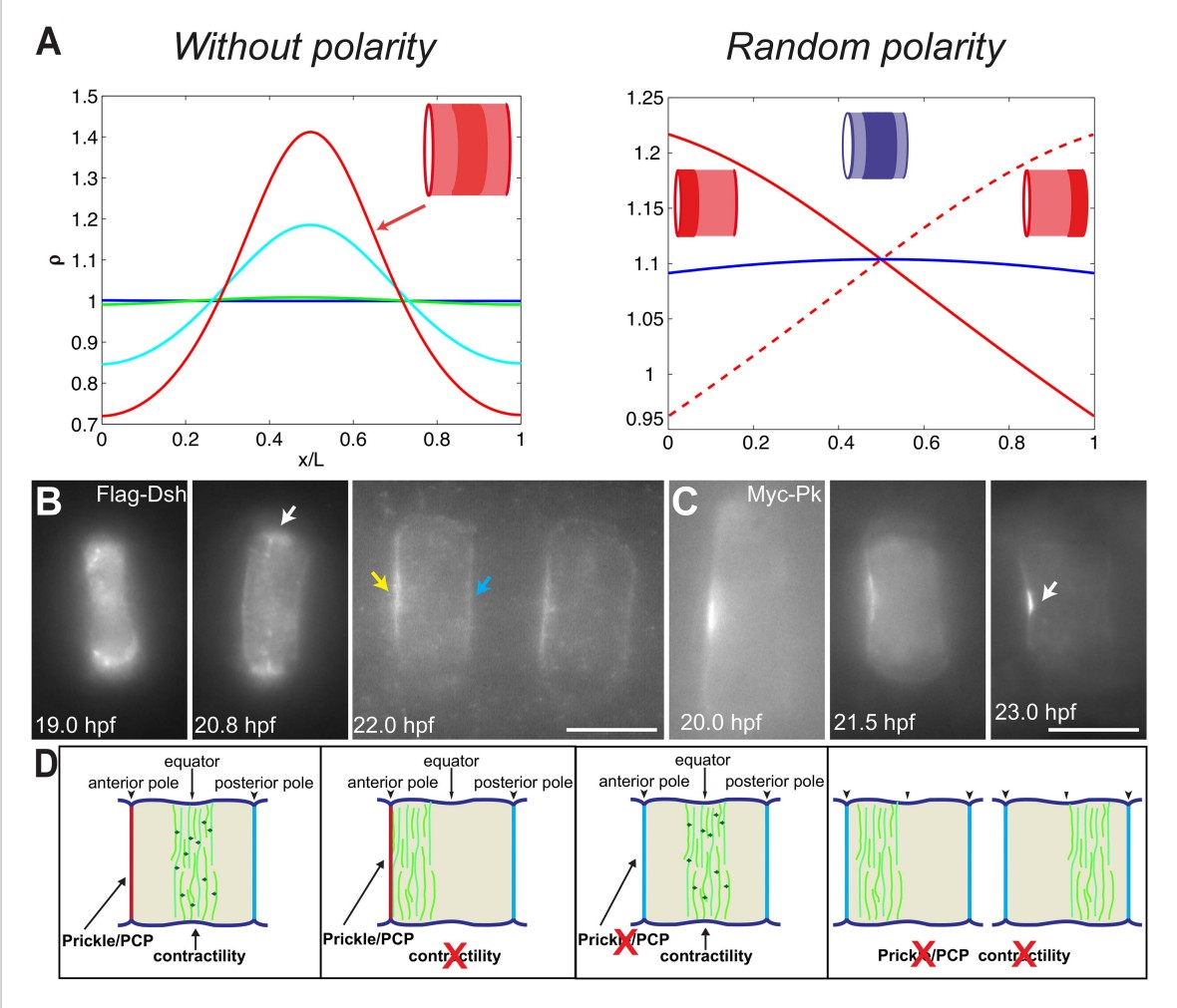

**Figure 7**. PCP participates in force balance to reposition actin rings. (**A**) Left: effect of a slow, 1.5-fold linear increase in contractility, for polarity-deficient mutants (no preferential flux on the edges). The ring forms directly at the center. Right: ring positioning for random uncoordinated polarity (preferential flux on the anterior, full red line, preferential flux on the posterior, dashed red line, and equal flux on anterior and posterior, full blue line). (**B**, **C**) Localization of Flag-Dsh (**B**), and Myc-Pk (**C**) in notochord cells. At early stages (19 and 20.8 hpf), Flag-Dsh localizes at the basal surface. At 20.8 hpf, it concentrates at the equator (white arrow). Subsequently, it shifts to both lateral surfaces, with a preference for anterior side of cells (yellow and blue arrows in **B**). Myc-Pk localizes at the anterior lateral surface of the cell at early stages and gradually concentrates to the center of anterior lateral surface (white arrow in **C**). (**D**) Myosin contractility antagonizes PCP to position a dynamic actin cytoskeleton. Anterior to the left. Scale bars, 10 μm.

Indeed, the equatorial ring in the notochord cells, despite its resemblance to the equatorial ring in cytokinesis, is established in a spindle-independent manner. A similar phenomenon has been reported in the asymmetrical cell division of *Drosophila* neuroblasts (*Cabernard et al., 2010*), which also uses cortical polarity signals to regulate the furrow.

During longer blebbistatin treatments, we observe complex oscillations of the cortex, with a second ring usually emerging at the equator. This suggests the existence of additional signals for ring formation and from a theoretical perspective, underlines the need to go beyond our simple one-component model, for instance, in order to take into account the dynamics of adhesion complexes as well. Indeed, our model is the simplest one can write for active gels, and it assumes for instance that all the rheological coefficients are constant, while in principle some, for example, viscosity, could depend on the state of the gel, and on blebbistatin concentration in the drug treatment (*Stirbat et al., 2013*). Nevertheless, the fact that we can capture well the experimental data with such a simple model suggests that such coupling effects are probably of secondary importance for the observed phenomenon. It should be noted that, in this study, we have considered polarity as a constant,

externally fixed parameter and examined its effect on actomyosin localization. In fact, a recent study in the same system (*Newman-Smith et al., 2015*) suggested an active feedback of myosin in core PCP protein localization. The next logical step would therefore be to incorporate this feedback loop to our formalism.

Earlier reports have noted movement of constrictions in isolated amphibian cells, especially the neural plate cells, which form local circumferential constriction rings traveling the length of the cells in successive waves (*Holtfreter, 1946*). Traveling constrictions have also been observed in leukocytes during cell shape changes and migration (*Senda et al., 1975*; *Haston and Shields, 1984*; *Shields and Haston, 1985*; *Bornens et al., 1989*).

Taken together, our data and theoretical analysis suggest a novel and generic framework through which PCP and contractility can interact to modify the forces acting on actomyosin rings and therefore their positioning (*Figure 7D*).

# Materials and methods

## Animals and fertilization

Adult *C. intestinalis* were obtained from Station Biologique de Roscoff, France and maintained in running filtered seawater. For fertilization, gametes from several individuals were surgically removed and mixed. Fertilized eggs were dechorionated with 1% sodium thioglycolate and 0.05% protease E as described by *Mita-Miyazawa et al. (1985)*, followed by five washes with UV-treated seawater. Embryos were cultured at 13°C.

Adult *C. savignyi* were collected at the Santa Barbara harbor (Santa Barbara, CA) and maintained in running seawater. Fertilization and dechorionation were performed as described above for *C. intestinalis*. For the *aimless* mutant, spawning was controlled by light conditions. Embryos were cultured at 15°C.

Adult *H. roretzi* were collected near the Asamushi Research Center for Marine Biology (Aomori, Japan) and the Otsuchi International Coastal Research Center (Otsuchi, Japan), and kept in tanks. Spawning was controlled by temperature and light conditions. Spawned eggs were fertilized with a suspension of non-self sperm. Embryos were cultured in Millipore-filtered seawater containing 50 µg/ml of streptomycin and kanamycin at 11°C.

## Plasmid constructs

Expression constructs in this study have been described previously: mCherry-UtrCH, lifeact-mEGFP, mCherry-MRLC and mCherry-hActin (*Dong et al., 2011*); mCherry-tropomyosin (*Sehring et al., 2014*); Flag-Dsh and Myc-Pk (*Jiang et al., 2005*).

## DNA electroporation

Electroporation was modified after previously published protocol (*Corbo et al., 1997*). Plasmid DNA (80 µg in 80 µl) was mixed with 400 µl 0.95 M mannitol in 4-mm cuvettes. 320 µl dechorionated fertilized eggs were added and electroporated with a Gene Pulser Xcell System (BIO-RAD), using a square pulse protocol (50 V and 15 ms per pulse). After electroporation, embryos were cultured at 13°C.

## Blebbistatin treatment

Blebbistatin dissolved in DMSO (Calbiochem, 203389) was used at a final concentration of 100 µM. Control embryos were treated with DMSO. For recovery experiments, embryos were washed 10× after blebbistatin treatment. All blebbistatin experiments were repeated at least three times.

## Phallacidin staining and immunohistochemistry

*C. savignyi* and *Halocynthia* embryos were fixed with 4% formaldehyde in seawater for 1 hr at room temperature (RT), washed 3 times with PBS, and stained with 5 units/ml BODIPY-FL phallacidin (Invitrogen, B607, Carlsbad, CA) in PBS containing 0.2% Triton X-100 for 2 hr at RT. After 3 washes for 10 min each with PBS, *C. savignyi* embryos were counterstained with DAPI, and transitioned through an isopropanol series with 30 s steps: 70%, 85%, 95%, and two times 100% isopropanol. For myosin staining in *Halocynthia*, fixed embryos were blocked with 0.1% BSA in PBT overnight at 4°C, followed by incubation with Rabbit anti Ser19 myosin antibodies (Cell Signaling Technology, 3671) (1:50) overnight at RT. After 2 × 40 min washes with PBT, embryos were incubated with Alexa594 anti-rabbit secondary antibodies (Invitrogen, A11011) overnight at RT, washed 3 times in PBT and counterstained

with 5 units/ml BODIPY-FL phallacidin in PBS. Localization of Dsh and Pk in *C. savignyi* followed previously published procedure (*Jiang et al., 2005*).

## Microscopy and image analysis

*C. intestinalis* embryos were observed under a Leica TCS SP5 confocal laser-scanning microscope (CLSM) equipped with a 40X oil-immersion objective (NA 1.2). If necessary, embryos were sedated using 0.2% MS222 (Sigma, A5040). *Halocynthia* embryos were analyzed with a BX61 CLSM (Olympus). *C. savignyi* embryos were imaged on an Olympus Fluoview 1000 CLSM using a $40 \times 1.3$ NA objective. Images were processed and analyzed with ImageJ. All fluorescent images shown are maximum projections. The ring width was measured on projections by drawing a line at the right angle to the lateral domains and across the ring. Kymographs were compiled using the kymograph plug in in ImageJ.

## Statistical analyses

Statistical parameters were determined using SigmaPlot software (Systat Software Inc.). Significance of differences was calculated using Student's *t* tests.

## FRAP

FRAP experiments were performed with the Leica TCS SP5. The whole cortical actin ring of notochord cells was bleached by using maximum laser power at 561 nm for an empirically determined number of iterations to achieve bleaching throughout the full thickness of the cortical signals. After bleaching, images were taken at regular intervals (between 4.5 and 9 s) with the same laser at 30% laser intensity. The halftime of recovery was calculated by measuring the signal intensity in the region of interest (ROI) over time. The raw data were corrected for background noise and image acquisition bleaching. The intensity was plotted as a function of time and the half-time of recovery, $t_{1/2}$, was extracted.

## Acknowledgements

We thank Drs Gaku Kumano and Hiroki Nishida for providing *Halocynthia roretzi* and laboratory spaces. This work was supported by grants 133335/V40 and 183302/S10 from the Norwegian Research Council to DJ and the NIH award GM088997 (MK). BD was supported by the projects ZR2015DM003 from NSF Shandong Province, 31572352 from NSFC, and Taishan Scholar Program of Shandong Province. EH was supported by a Young Investigator Award from the Bettencourt-Schueller fundation.

## Additional information

### Funding

| Funder | Grant reference | Author |
|---|---|---|
| Norges Forskningsråd | 133335/V40 and 183302/S10 | Di Jiang |
| National Institutes of Health (NIH) | GM088997 | Matthew Kourakis |
| National Science Foundation (NSF) of Shandong province and China | ZR2015DM003 and 31572352 | Bo Dong |

The funders had no role in study design, data collection and interpretation, or the decision to submit the work for publication.

### Author contributions

IMS, DJ, Conception and design, Acquisition of data, Analysis and interpretation of data, Drafting or revising the article; PR, EH, Acquisition of data, Analysis and interpretation of data, Drafting or revising the article; ED, Acquisition of data, Analysis and interpretation of data; MK, Acquisition of data, Drafting or revising the article; BM, Acquisition of data, Contributed unpublished essential data or reagents; BD, Analysis and interpretation of data, Drafting or revising the article

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

## Appendix 1

## Physical modeling of the *Ciona* cortical flows

This text contains technical derivations regarding the model of ring migration as well as quantitative connections with the experiments.

### The model

We model the cell cortex as a thin cylindrical layer of an actomyosin gel of length $L$ with principal axis $z \in [-L/2, L/2]$. In order to capture the most basic aspects of ring formation and positioning, we use a one-dimensional description and do not resolve the radial direction which has been studied in (**Joanny et al., 2013**). Therefore, all equations are invariant in the orthoradial direction. We will start by considering separately actin filaments and monomers and attached myosin motors, viewing the motors as being in solution in the actin solute.

We denote by $a(z,t)$ the concentration of actin filaments, by $m(z,t)$ the concentration of actin monomers and by $c(z,t)$ the concentration of motors (or to be more accurate bipolar filaments including 20–30 motors) performing power stroke on the actin filaments. Following (**Kruse et al., 2005**), the conservation equations then read,

$$\begin{aligned} \partial_t a + \partial_z(av) &= k_p m - k_d a \\ \partial_t m - D_m \partial_{zz} m &= -k_p m + k_d a \\ \partial_t c + \partial_z(cv) - D_c \partial_{zz} c &= k_{on} - k_{off} c \end{aligned} \qquad (1)$$

The velocity of actin filaments is $v(z,t)$. Actin polymerizes (resp. depolymerizes) with fixed rates $k_p$ (resp. $k_d$), and monomers are free to diffuse in the cytoplasm with a coefficient $D_m$. We assume that motors are advected with actin but can also thermally diffuse with a diffusion coefficient $D_c$. Microscopic derivation from a two state motor (**Julicher et al., 1997**) of such type of macroscopic equation is given in **Zeldovich et al. (2005)**. On top of this, motors can also attach and detach away from the actin filaments again following a first order kinetic which involves rates $k_{on}$ and $k_{off}$. We also suppose that there is a large pool of free motors.

To begin with, we assume that the polymerization/depolymerization reaction is close to equilibrium such that can write:

$$m = Ka, \qquad (2)$$

where $K$ is the reaction constant. Combining the first and second equation of (**Equation 1**), we obtain:

$$(1 + K)\partial_t a + \partial_z(av) - KD_m \partial_{zz} a = 0. \qquad (3)$$

Supposing that $K \ll 1$ while $D_a = KD_m$ remains finite, we have $\partial_t a + \partial_z(av) - D_a \partial_{zz} a = 0$.

Thus, $D_a$ is an effective diffusion of actin filaments that accounts for actin filament production and degradation. We rewrite **Equation (1)** as a set a reaction-advection-diffusion equations.

$$\begin{aligned} \partial_t a + \partial_z(av) - D_a \partial_{zz} a &= 0 \\ \partial_t c + \partial_z(cv) - D_c \partial_{zz} c &= k_{on} - k_{off} c \end{aligned} \qquad (4)$$

Added to this, we assume that the friction forces stemming from the membrane and the permeation friction forces (**Rubinstein et al., 2009**; **Joanny et al., 2013**) with the cytosol are negligible. To check this hypothesis, we compared experimentally the anterior and posterior angles between lateral and basal sides ($n = 10$), around 19–20 hpf, during ring migration towards the center. If friction was significant, these angles would be expected to be different for an anterior-positioned ring, as basal tension would not be constant throughout the basal cortex (more precisely, the difference would be $\sigma(L/2) - \sigma(-L/2) = \xi \int_{-L/2}^{L/2} v(z)\mathrm{d}z$ where $\xi$ is the friction coefficient of the cortex against the extra-cellular matrix, and $\sigma$ is the internal stress in the

actomyosin network). Nevertheless, we could not find a statistically significant difference between these two angles. Thus, force balance reads,

$$\partial_z \sigma = 0. \tag{5}$$

and $\sigma \equiv \bar{\sigma}$ is the homogeneous stress inside the acto-myosin cortex which is imposed by neighbouring cells.

Finally, the behaviour law for stress in the actomyosin network is given by **Julicher et al. (2007)**,

$$\bar{\sigma} = \eta \partial_z v + \chi^0 a^2 m - Ea, \tag{6}$$

where $\eta$ is the viscosity of actomyosin, and $E$ is a compressibility modulus. We have assumed a simple model for the active stress, where one myosin motor needs to interact with two actin filaments to perform a power stroke, with a positive contractility coefficient $\chi^0$.

To further simplify our description, we effectively treat actin and myosin as a single species of density $\rho$ of acto-myosin satisfying a reaction-drift-diffusion equation (**Peskin et al., 1993**; **Mogilner and Oster, 1996**; **George et al., 2013**; **Dierkes et al., 2014**):

$$\partial_t \rho + \partial_z (\rho v) - D \partial_{zz} \rho = \frac{\rho_0 - \rho}{\tau}, \tag{7}$$

where $D$ is an effective diffusion coefficient of acto-myosin, $\rho_0$ a target density, and $\tau$ a turnover time. To linear order, the behavior law simply becomes,

$$\bar{\sigma} = \eta \partial_z v + \chi \rho, \tag{8}$$

where $\chi$ is the effective contractility coefficient. In the main text, we show that $\partial_z v$ decays linearly with actin fluorescence intensity. This further indicates that $\bar{\sigma}$ in **Equation (8)** can indeed be considered as constant and the friction coefficient with the extra-cellular matrix can indeed be considered negligible.

Notice that our **Equations (7, 8)** are generic in the sense they may be obtained with a slightly different interpretation. Starting from (4)$_2$, we may directly write the contraction term in the behavior law (6) in a linear way $\chi c$ as done in (**Bois et al. (2011**; **Wolgemuth et al., 2011**) and thus obtain the same set of equation where the concentration field is the one of motors instead of acto-myosin. The turnover time $1/k_{off}$ thus represents the one of attachment–detachment of motors which is experimentally found to be close to the one of actin. Assuming infinite compressibility, the actin concentration uncouples from the rest of the problem and can be reconstructed post-factum from (4)$_1$ as done in **Recho et al. (2013**, **2014)** once the velocity field is known.

Finally, we turn to boundary conditions. Actomyosin can be produced locally at the anterior and posterior boundaries. Thus denoting the flux,

$$J = \rho v - D \partial_z \rho, \tag{9}$$

we have,

$$J(-L/2) = J_a \quad \text{and} \quad J(L/2) = J_p. \tag{10}$$

Notice that this flux contains the effective diffusion coefficient stemming from actin filaments turnover. Thus, PIV does measure the velocity of filaments $v$ and not the flux $J$ which is related to mass balance.

One mechanical condition is still missing because the system is friction free, so all rigid body motions are allowed. To eliminate these modes, we additionally set:

$$\int_{-L/2}^{L/2} v(z, t) \, dz = 0. \tag{11}$$

It is in agreement with imposing that the stress is the same at both sides in the presence of friction (integration of $\partial_z \sigma = \xi v$).

## Appendix 2

### Rescaling

Non-dimensionalizing spacial variables by $\sqrt{D\tau}$, time by $\tau$, stress by $\eta/\tau$ and density by $\rho_0$, denoting $x = z/L \in [-1/2, 1/2]$, we get from **Equations (7, 8)**,

$$\mathscr{L}^{-1}\partial_x v = \mathscr{S} - \mathscr{C}\rho$$

$$\mathscr{L}^{-1}\partial_t(\mathscr{L}\rho) + \mathscr{L}^{-1}\partial_x J = 1 - \rho \quad , \tag{12}$$

$$\mathscr{L}^{-1}\partial_x \rho = \rho\left(v - x\frac{d\mathscr{L}}{dt}\right) - J$$

with boundary conditions,

$$J(-1/2) = \mathscr{J}_a, \ J(1/2) = \mathscr{J}_p \ \text{ and } \int_{-1/2}^{1/2} v(x)\,dx = 0. \tag{13}$$

The five non-dimensional parameters entering the problem are, the contractility, $\mathscr{C} = \chi\rho_0\tau/\eta$, the residual stress $\mathscr{S} = \bar{\sigma}\tau/\eta$, the normalized length of the cell, $\mathscr{L} = L/\sqrt{D\tau}$ and the fluxes $\mathscr{J}_{a,p} = J_{a,p}/\rho_0\sqrt{\tau/D}$. After these five parameters and an initial data is provided, system (**Equation 12**) can be solved numerically using a finite volume discretization stencil (**LeVeque, 2002**).

### Assumptions

We set the posterior actin flux to zero considering that the anterior flux is much larger at least in the control case. The robustness of such assumption will be numerically checked.

Added to this, as an homogeneous steady state $\rho \equiv 1$ is expected in the absence of anterior flux for low contractilities, we necessarily have to set $\mathscr{S} = \mathscr{C}$ to ensure such steady state. Physically, this condition states that the external stress $\bar{\sigma}$ applied by neighboring cells balances the active contractile stress $\chi\rho_0$ to maintain the shape.

## Appendix 3

### Steady states

At steady state, density does not vary in time ($\partial_t \rho \equiv 0$) and length increase also vanishes ($d\mathscr{L}/dt = 0$). The flux of actomyosin then satisfies,

$$-\mathscr{L}^{-2}\partial_{xx}J = \mathscr{C}\left(1 - \mathscr{L}^{-1}\partial_x J\right)\left(J - \int_{-1/2}^{1/2} J(u)\,du\right) - J \tag{14}$$

with $J(-1/2) = \mathscr{J}_a$, and $J(1/2) = 0$. **Equation (14)** has a long range interaction with a quadratic non linearity structure.

### No flux case

When $\mathscr{J}_a = 0$, $J \equiv 0$ is a trivial solution of **Equation (14)**. However, an increase of contractility can destabilize this homogeneous configuration which stops to be stable when the critical contractility threshold,

$$\mathscr{C}_c = 1 + 4\mathscr{L}^{-2}\pi^2, \tag{15}$$

is reached. At $\mathscr{C}_c$, the stable solution bifurcates from the homogeneous one in a pitchfork supercritical way (second order phase transition) and we have

$$\sqrt{\int_{-1/2}^{1/2} J(x)^2 dx} \underset{\mathscr{C}_c}{\sim} \pm\sqrt{\frac{2\sqrt{2}(\mathscr{C} - \mathscr{C}_c)}{\mathscr{C}_c^2}}. \tag{16}$$

The bifurcation eigenmode is thus

$$J_c(x) \underset{\mathscr{C}_c}{\sim} \pm\sqrt{\frac{2\sqrt{2}(\mathscr{C} - \mathscr{C}_c)}{\mathscr{C}_c^2}}\sin(2\pi(x + 1/2)), \tag{17}$$

which concentrates actin either in the center or on the edges depending on the prefactor sign.

Notice that upon increase of $\mathscr{C}$, there is a infinite number of bifurcation branches branching at points $1 + 4n^2\mathscr{L}^{-2}\pi^2$ where $n \geq 2$; however, all these branches were numerically found to be unstable. So the situation is that if $\mathscr{C} < \mathscr{C}_c$, the homogeneous solution is the global attractor while when $\mathscr{C} > \mathscr{C}_c$, the global attractor is the solution localizing (more and more with increase of $\mathscr{C}$) to the center or the poles (see **Appendix figure 1**)

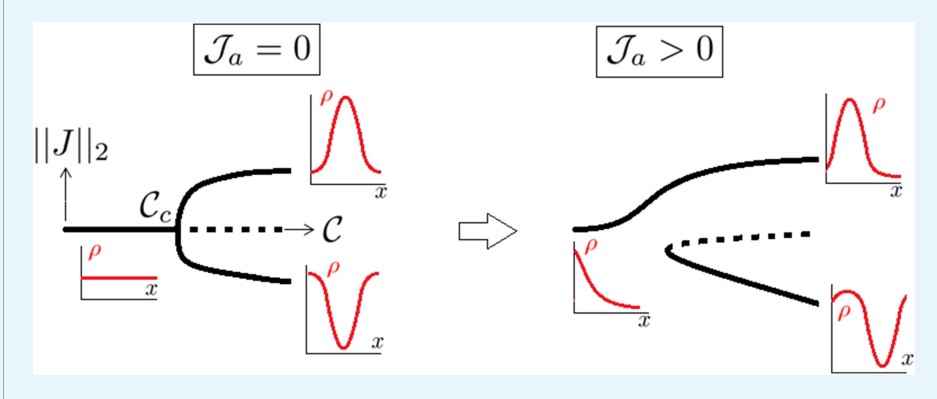

**Appendix figure 1**. A supercritical pitchfork bifurcation is broken into a saddle one upon introduction of a non symmetrical perturbation. Dotted lines indicate that the branch is linearly unstable.

## Influence of the anterior flux

Introducing the anterior mass flux brings a bias inside the problem and breaks the pitchfork bifurcation into a saddle node one as generically explained in **Arnold et al. (1994)** (**Appendix figure 1**).

The biasing favors the formation of an actin ring at the anterior pole. Indeed, one can solve **Equation 14**, when $\mathscr{C}=0$, to find,

$$J_{\mathscr{C}_0}(x) = \mathscr{J}_a \frac{\sinh\left(\mathscr{L}\left(\frac{1}{2}-x\right)\right)}{\sinh(\mathscr{L})}. \tag{18}$$

and thus,

$$\rho_{\mathscr{C}_0}(x) = 1 + \mathscr{J}_a \frac{\cosh\left(\mathscr{L}\left(\frac{1}{2}-x\right)\right)}{\sinh(\mathscr{L})}. \tag{19}$$

The density of actin clearly localizes to the anterior pole over a characteristic length $\mathscr{L}$.

In the general case, upon increase of contractility, the distribution then switches from localization at the anterior pole when $\mathscr{C}=0$ to a localization close to the center when $\mathscr{C}$ is large.

Notice that the branch localizing the motors on both poles is still locally stable. However, it is disconnected from the other one where there is a switch from the side to the center. Also we numerically find that the basin of attraction of this branch is getting smaller as $\mathscr{J}_a$ increase. As a result, it is probably difficult to observe experimentally even when polarity is impaired.

## Appendix 4

## Rough estimates of model parameters

To begin with, we estimated the typical scales of time and space.
- FRAP experiments allow to estimate $\tau = 90$ s.
- An approximation of the effective diffusion of acto-myosin can be found from (**Mogilner and Oster, 1996**; **Mogilner and Edelstein-Keshet, 2002**). The diffusion coefficient of actin monomers in the cytoplasm is 20 $\mu m^2$ $s^{-1}$ and $k_d/k_p \approx 1/100$. This leads to $D \approx 0.2$ $\mu m^2$ $s^{-1}$. We then obtain,

$$\sqrt{D\tau} \approx 3.16 \ \mu m.$$

We estimate below the non dimensional parameters of the model.

### Parameter $\mathscr{L}$

From experiments, averaging over a number of 7 cells, we estimate the length of cell to increase during the ring migration from $L \approx 7$ $\mu m$ to $L \approx 12$ $\mu m$. This implies for the non dimensional length an increase from $\mathscr{L} = 1.6$ to $\mathscr{L} = 2.8$. As we show on **Appendix figure 2**, the trend of the stretch (defined as the current length over the final length) is roughly linear and for simplicity, we implement this trend in the numerical simulations of the theoretical model. After the migration, length can be considered to be roughly constant. We show of **Appendix figure 3** the evolution of length after a blebbistatin treatment at time = 0 min.

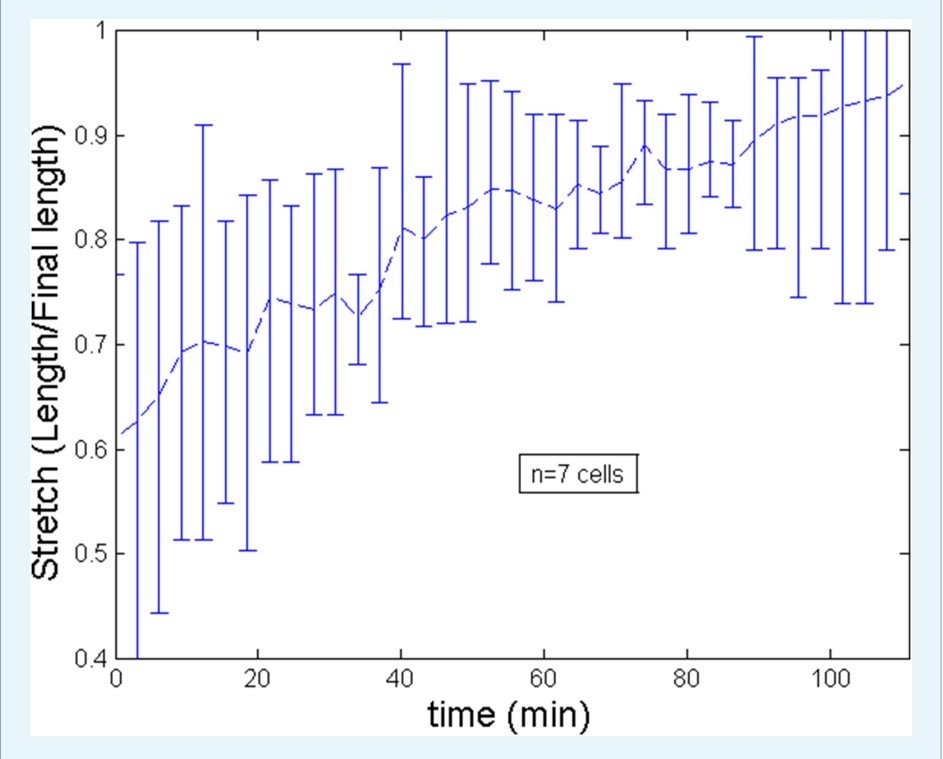

**Appendix figure 2**. Stretch of the cell during the ring migration.

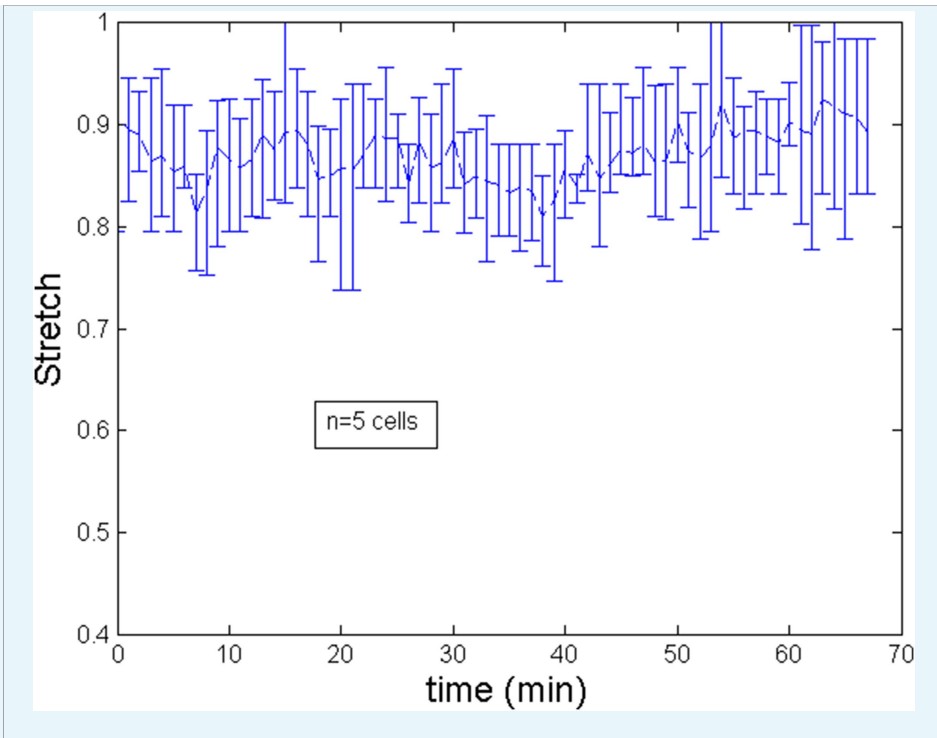

**Appendix figure 3**. Stretch of the cell during after ring migration subsequent to a blebbistatin treatment.

## Parameter $\mathscr{I}_a$

PIV as well as kymographs reveals that the average velocity of actin filaments is $v \approx 30$ nm s$^{-1}$.
$\mathscr{I}_a/\rho_0$ should therefore have a comparable order of magnitude. Using estimates of $\tau$ and $D$, this
leads to $\mathscr{I}_a \approx 0.6$. To fit the data, we take a slightly smaller value $\mathscr{I}_a \approx 0.4$ which is also
confirmed by the intensity profiles after blebbistatin treatment which value at the anterior side
is $1 + \mathscr{I}_a$ (see **Equation 19**).

## Parameter $\mathscr{C}$

We begin with a estimate of $\mathscr{C}$ when the ring is formed. From relation (**Equation 8**), we have
that,

$$\eta \frac{\Delta v}{\Delta L} \sim \chi(\rho - \rho_0).$$

Maximal velocity estimated by PIV is of the order of $v \approx 50$ nm s$^{-1}$ and decays to zero over
roughly a fourth of the cell length that is 3 μm. Over the same length, intensity of actin
fluorescence can increase roughly of one half of its base value leading $\rho - \rho_0 \sim 0.5\rho_0$. We thus
obtain $(\chi\rho_0)/\eta \approx 0.03$ s$^{-1}$. This ratio is in agreement with the estimates of (**Julicher et al., 2007**;
**Rubinstein et al., 2009**), which suggest $\chi\rho_0 \approx 10^3$ Pa, $\eta \approx 10^4$–$10^5$ Pa · s. We then obtain $\mathscr{C} \approx 3$.
For the numerical model, to fit the experimental data, we postulated that $C$ increases linearly in
time from $\mathscr{C} = 2.1$ to $\mathscr{C} = 3.3$ thus assuming an increase of the (homogeneous) cortical tension
of roughly 1.5 as measured experimentally during the course of development (see **Appendix
figure 5B** of the main text where an elongation of 72% is related to an increase of the rescaled
tension from 1.2 to 1.8, hence the assumed 1.5 increase). Notice that this is this increase that
dictates the ring migration dynamic. Indeed, from the average filament velocity $v \approx 30$ nm s$^{-1}$,
it would take ~3 min for the ring to migrate from the side to center position (a ~6 μm distance)
if the internal flow were driving the ring positioning.

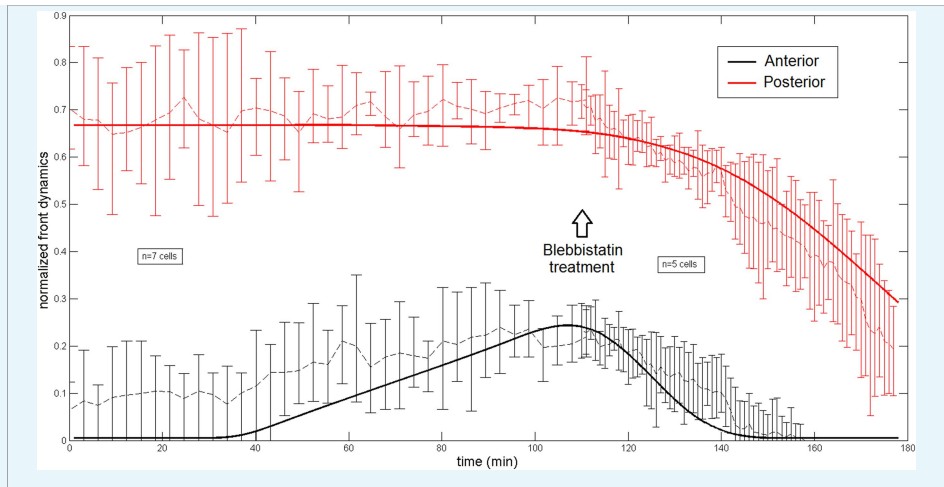

**Appendix figure 4**. Comparison of the model and experimental anterior and posterior lateral domains during normal development blebbistatin treatment.

After blebbsitatin treatment, we assume that $\mathscr{C}$ exponentially decreases to a value $\mathscr{C}_{bleb}=0$ after a characteristic time $\tau_b = 15$ min. The effect of the blebbistatin dose has not been quantified experimentally, and we shall present its effect by changing $\mathscr{C}_{bleb}$ in the robustness section.

## Appendix 5

# Model predictions

Having estimated the parameters of the model, we turn to its quantitative predictions and compare numerical results of the model simulation with experiments.

## Development of normal embryos followed by blebbistatin treatment

For the control case, as mentioned in the previous section, we suppose that in the course of ring migration which average (over 7 cells) duration is ~110 min, $\mathscr{C}$ increases linearly from 2.1 to 3.3 while $\mathscr{L}$ increases linearly from 1.6 to 2.8. After the ring forms, we apply a blebbistatin treatment and predict a migration back to the anterior pole. The average time of such migration (over 5 cells) is 67 min. During this repositioning process, we assume accordingly with experimental data that the cell length remains roughly constant (see **Appendix figure 3**) while $\mathscr{C}$ drops exponentially over 15 min. We start the simulation from an initial actin density distribution with no contractility (**Equation 19**) and iterate from few time steps keeping initial parameters constant to obtain the real starting profile for non zero $\mathscr{C}$.

### Dynamics of the anterior and posterior domains

We show on **Appendix figure 4** the model and experimental anterior and posterior lateral domains dynamics which are in very good agreement. The theoretical lateral domains are defined by considering the domain where the actin intensity is superior to the minimal value plus 50% of difference between maximal and minimal intensity. Distances are normalized by the total cell length for a more convenient representation. Here, 0 corresponds to the cell anterior while 1 is the cell posterior. Both the anterior and posterior lateral domains are well captured by the model during the ring migration of normal embryos as well as after the ring repositioning subsequent to a blebbistatin treatment happening at the end of the migration at a time = 110 min. Notice that accordingly with experiments, the location of the ring at the end of the migration at time = 110 is systematically slightly shifted to the anterior side, revealing the essential asymmetry between the two sides which we interpret in the model in term of an anterior production flux. The ring shifts back to the anterior pole at a velocity much higher the original one as observed experimentally.

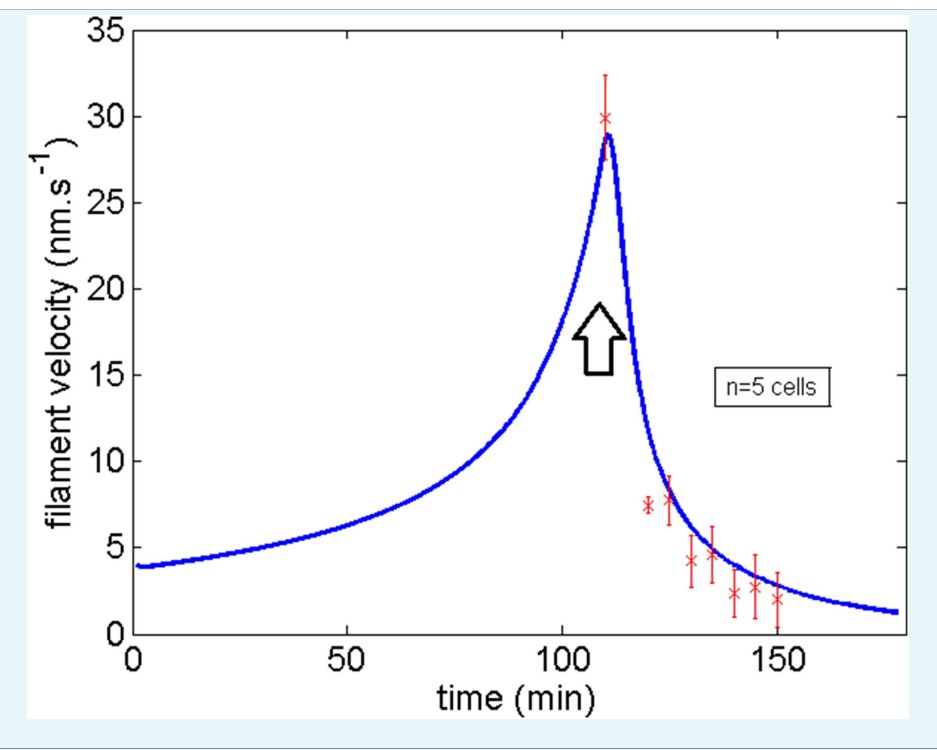

**Appendix figure 5**. Comparison of the model and experimental filament average velocity during development and blebbistatin repositioning. Black arrow indicates the blebbistatin treatment.

## Average value of the actin filaments velocity

The average value of the actin filaments velocity is computed from experiments by summing absolute values of velocities at several locations of the ring during the ring relocation after blebbistatin treatment. The value,

$$\overline{V} = \frac{1}{L} \int_0^L |v(z,t)| dz,$$

obtained from the model is also monitored and compared against available experimental on **Appendix figure 5**.

The agreement is again quite remarkable for the velocity of the actin filaments at the end of migration (29 nm s$^{-1}$) and the sharp decrease of the filament velocity upon blebbistatin treatment is nicely captured.

## Scaling of the ring size

Another prediction of the model in agreement with experiments is the scaling of ring size with the cell size up to a threshold length. See **Appendix figure 6C** of the main text and **Appendix figure 6**.

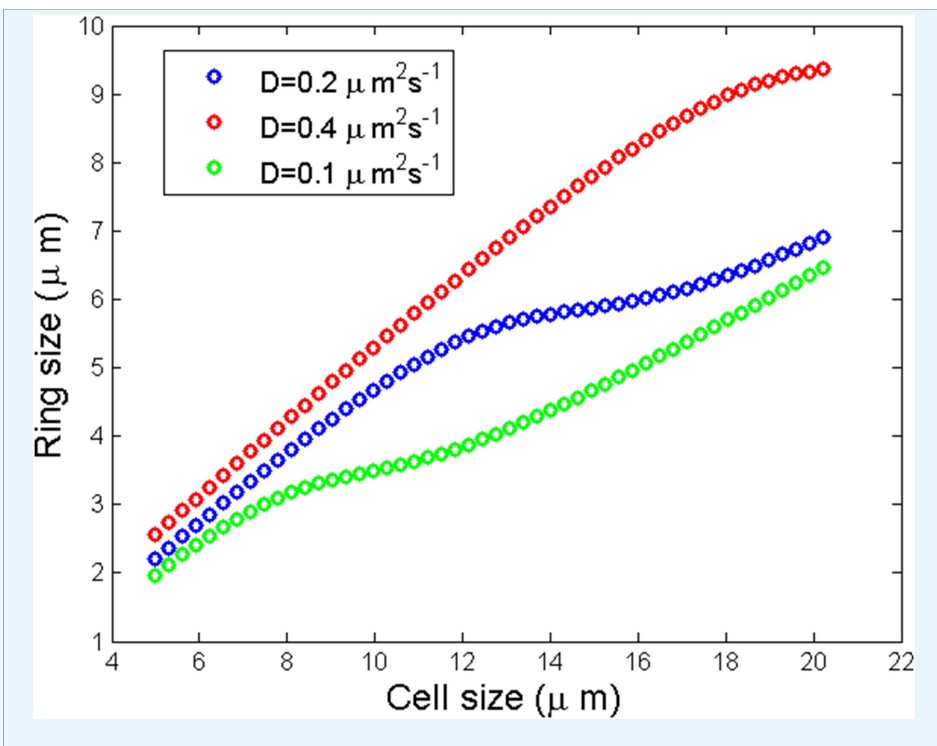

**Appendix figure 6**. Numerical simulation of ring size as a function of the cell size for three different diffusion coefficients.

We observed a linear scaling of the ring size with respect to the cell size followed by a saturation and a linear increase again. The qualitative reason of the saturation can be understood for small enough $\mathscr{I}_a$ where one may speculate that the saturation size corresponds to the critical cell size:

$$L_c = 2\pi\sqrt{\frac{D\tau}{\mathscr{C}-1}}.$$

## Intensity and velocity profiles

We show on **Appendix figure 6D** and **Appendix figure 6F** of the main text, a comparison of the experimental and theoretical density of actin filaments for several time frames during the course of normal migration and blebbistatin treatment. The profiles of fluorescence obtained experimentally compare well to the theoretical one.

Added to this, we also show on **Appendix figure 5** of the main text the velocity found by PIV analysis at the end of migration which shape is very similar to the one predicted by the model. The maximal velocity of actin filaments (50 nm s⁻¹) found theoretically is also almost the same as the experimentally measured ones. This validates post factum the procedure we used to assign values to non dimensional parameters explained in the previous section.

## Robustness analysis

We now check the robustness of our results with respect to few hypothesis.

## Posterior flux

Firstly, we show that the anterior and posterior domains dynamic are not importantly altered (see **Appendix figure 7**) if we add a small posterior flux $\mathscr{I}_p = -0.04$ ten times lower than $\mathscr{I}_a = 0.4$. Generically, such a non-zero flux could be expected. In fact, the agreement with the experimental ring dynamic is even better in such case and we may speculate that such flux indeed exists.

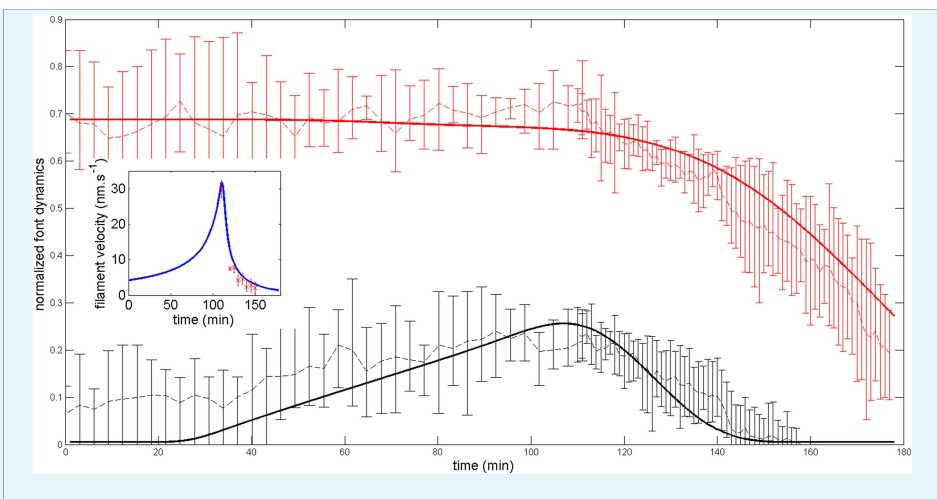

**Appendix figure 7**. Robustness of the anterior and posterior fronts dynamics and average filament velocity during the ring migration in control case with $\mathscr{I}_p = -0.04$.

## Boundary tension

The neighboring cells may not always apply a tension such that $\mathscr{S} = \mathscr{C}$ though this choice is the most natural one as it ensures a homogeneous cortex if the active processes of maintaining contraction and polarity are turned off.

If we suppose that $\mathscr{S}$ is less than $\mathscr{C}$ leading to the condition $\mathscr{S} = \alpha\mathscr{C}$ where $\alpha < 1$, we show below the effect of a small deviation $\alpha = 0.95$. See **Appendix figure 8**. While the anterior and posterior dynamics is still correctly captured, the agreement of the average filament velocity is not good any more even though the order of magnitude is still correct. The system is thus sensitive to this condition. However, if the condition is modified by assuming a certain value of $\alpha$, by changing other parameters of the model, we can still obtain a good agreement with the experimental data. What this analysis suggests, if the model is correct, is that a change of externally applied tension would strongly affect the internal cortical flow, a fact which awaits experimental verification in the present setting.

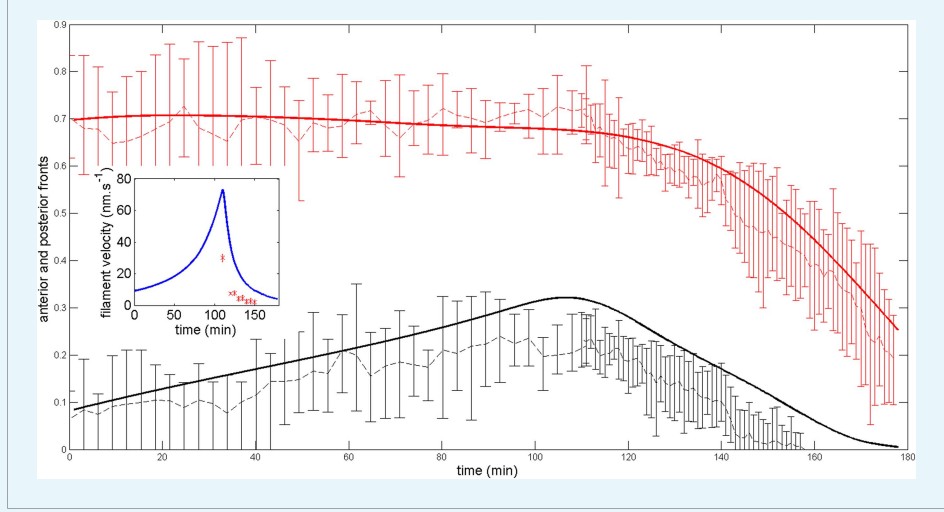

**Appendix figure 8**. Robustness of the anterior and posterior fronts dynamics and average filament velocity during the ring migration in control case with $\mathscr{S} = 0.95\,\mathscr{C}$.

## Blebbistatin dose dependence

To mimic the fact that the dynamic of the shift is blebbistatin dose dependent, we assume that after the ring has migrated, contractility $\mathscr{C}$ decreases to a value $\mathscr{C}_{bleb} > 0$. We show on **Appendix figure 9** how the dynamic of the shift back to the anterior pole is affected. A more modest impairment of contractility slows down the ring migration velocity as observed experimentally.

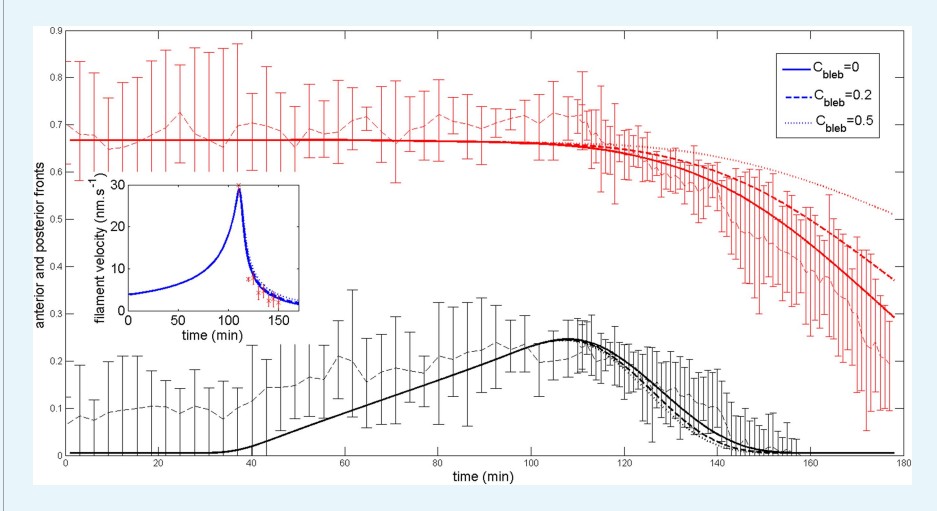

**Appendix figure 9**. Effect of a blebbistatin treatment at time = 110 min on the ring dynamic with $\mathscr{C}_{bleb} = 0, 0.2, 0.5$.

## Development of aimless embryos followed by blebbistatin treatment

For aimless embryos, $\mathscr{J}_a$ is considerably lowered and has the same order of magnitude as $\mathscr{J}_b$. If the two fluxes are exactly zero, when contractility increases above $\mathscr{C}_c \approx 6$, starting from an almost homogeneous distribution of actin, the model predicts the formation of a central ring directly in the center. The configuration where actin is localized at the poles is most probably never observed because of the small remaining anterior and posterior actin fluxes. As a consequence of such fluxes, the threshold of appearance of the ring is also not exactly $\mathscr{C}_c$.

To model the dynamic of this situation, we consider small stochastic actin fluxes of the order of $\mathscr{J}_a \sim -\mathscr{J}_p \sim 0.04$. The dynamic of length and contractility remains unchanged from the control case. We show on **Appendix figure 10** the resulting anterior and posterior lateral domains positions and average filaments velocity. Observe that as $\mathscr{J}_a \sim \mathscr{J}_p$ are small, we now start from a nearly homogeneous density. When contractility increases, a ring forms in the middle of the cell as observed experimentally. The duration of the ring formation remains unchanged as it is driven by the contractility increase. However, the average filament velocity is considerably lowered. If $\mathscr{J}_a$ and $\mathscr{J}_p$ are strictly the same, the dynamic of the anterior and posterior fronts are symmetric but a small imbalance (either to the posterior or anterior side) is sufficient to trigger a side to middle migration as in **Appendix figure 10** though the density of actin is much more homogeneous then in the control case.

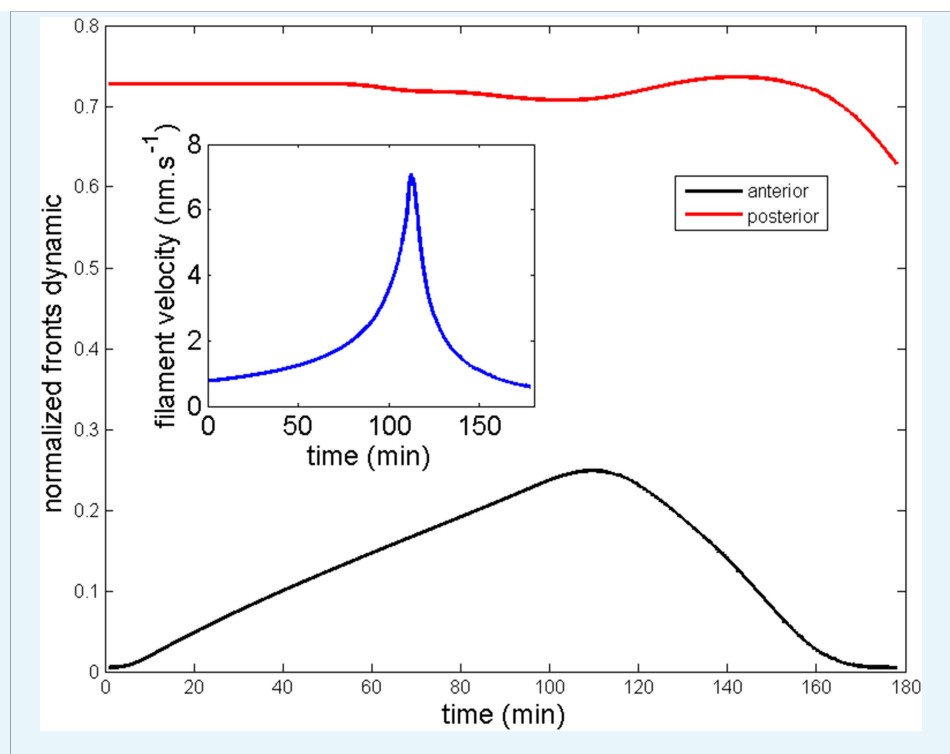

**Appendix figure 10**. Ring migration in the aimless case. Parameters are $\mathscr{I}_a = 0.04$ and $\mathscr{I}_p = -0.02$.

After a time of 110 min, depending on the importance of the impairment of contractility or polarity, the ring may shift to the anterior or posterior pole depending on which (anterior or posterior) flux dominates the other. In most cases, both anterior and posterior fluxes will be of about the same magnitude and the ring central position (defined as the middle of the anterior and posterior lateral domains) will thus stay in the center. To illustrate this case, we show on **Appendix figure 11**, three typical cases with blebbistatin treatment ($\mathscr{C}_{bleb} = 1$) corresponding to $\mathscr{I}_a = 0.04$ and $\mathscr{I}_p = -0.02$, $\mathscr{I}_a = 0.02$ and $\mathscr{I}_p = -0.04$ and finally $\mathscr{I}_a = -\mathscr{I}_p = 0.04$.

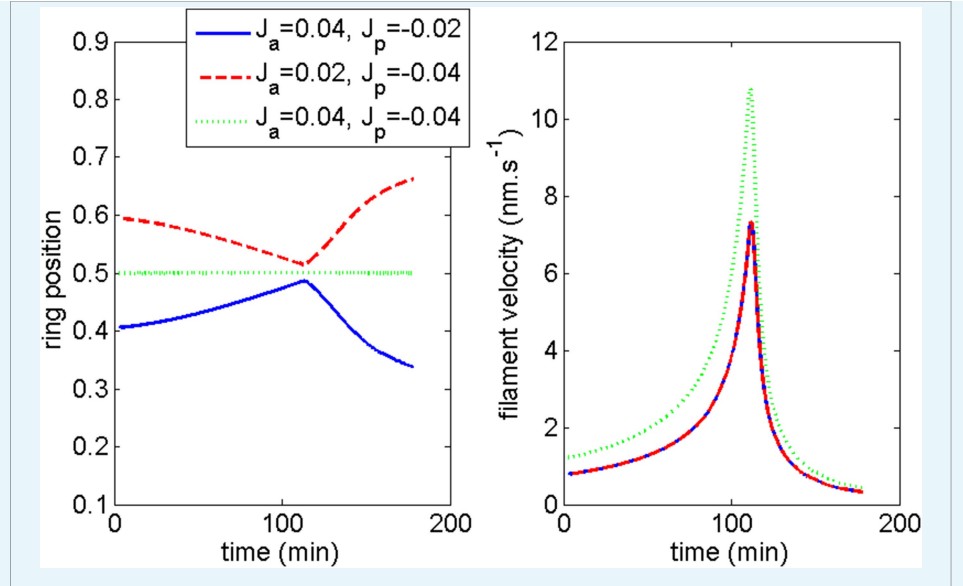

**Appendix figure 11**. Ring middle position and average filament velocity during migration of the ring and after blebbistatin treatment at time = 110 min in the three cases mentioned in the text.

In agreement with experiments, in all three cases, the ring migrates towards the middle of the cell during development. Then, after blebbistatin treatment, depending over which flux slightly dominates, the ring re-positions in the central region or at one of the sides randomly.

