## [Decision Letter]

[Editors’ note: a previous version of this study was rejected after peer review, but the authors submitted for reconsideration. The previous decision letter after peer review is shown below.]

Thank you for choosing to send your work entitled “Assembly and positioning of actomyosin rings by contractility and PCP” for consideration at *eLife*. Your full submission has been evaluated by Vivek Malhotra (Senior Editor), a Reviewing Editor and three peer reviewers, and the decision was reached after discussions between the reviewers. Based on our discussions and the individual reviews below, we regret to inform you that your work will not be considered further for publication in *eLife*.

Dr. William Bement (Reviewer 1) and Dr. Mohan Balasubramanian (Reviewing Editor) have agreed to reveal their identity.

All referees find your experimental observations exciting and the data of high quality. However, 2 of the 3 referees strongly feel that the modelling studies neither makes non-trivial predictions nor tests hypothesis generated (for the modeling to be useful). Given that the paper includes modeling and experiments, we are therefore compelled to look at both parts in totality. Therefore, we cannot offer to take this manuscript further in its current form.

Since the area of research and the observations are of interest to all referees (and potentially to the readership of *eLife* as well), we remain interested in this work and area. However, the revisions required are extensive and it is not clear what form the revised manuscript may take. If you can address the comments of the referees satisfactorily, we will be happy to consider this paper further (as a new submission that will be sent to the same referees). If you want to take this route, please be sure to address all points, but importantly strengthen the modeling part with validation if at all possible.

*Reviewer #1*:

It could be reasonably argued that in spite of all of the triumphs that have accrued to developmental biology as result of genetic and molecular analysis of morphogenesis, one major question remains unaddressed: how, exactly, molecules are tied to cell shape changes. On the one hand, an enormous body of information concerning developmentally important transcription factors and their target genes exists and on the other hand, cell biology has provided a wealth of information about the primary agent of cell shape change – the actomyosin cytoskeleton. And yet the connections between the two remain maddeningly elusive.

In the current study, the authors provide a fascinating look into one such connection – that responsible for driving elongation of notochord cells of an ascidian. In brief, the authors show that an equatorial ring of actin filaments and myosin-2, previously found to be responsible for powering elongation, assembles as a result of myosin-2 powered contraction and resultant cortical flow of F-actin. Initially, F-actin is concentrated in a ring at the anterior end of the cells, but it moves parallel to the plane of the cortex to the middle of the cell, where it is needed to do its job. Myosin-2 shows a similar behavior and disruption of its motor activity with blebbistatin (which blocks elongation) prevents repositioning of the actin ring. Moreover, blebbistatin causes repositioning (to the anterior) of the ring when applied after the achievement of its central location. The authors develop a simple model, which successfully captures some of the key features observed, but does one essential feature – the starting localization at the anterior of the cell. This prompted a series of experiments designed to test the possibility that the planar cell polarity (PCP) pathway is responsible for the starting localization. They found that it is, providing a link between a pathway implicated in several different morphogenetic events (PCP) and a cellular process that actually powers a morphogenetic event. The authors modified their model which was thereupon able to capture all of the relevant results.

This is a very nice piece of work, not only for the reasons presented above, but also for the quality of the data. The authors obviously appreciate the importance of high quality imaging and also obviously appreciate what to do with the results provided by it. I thus have just a couple of comments:

1) Based both on the movies and on the fact that many F-actin nucleators concentrate at cell-cell junctions, I was puzzled by the assumption that “actin fluxes vanish” at cell-cell junctions (Results) underlying the model. Am I just misunderstanding what is meant by flux in this context? Perhaps some clarification is in order, as it seems manifestly obvious that F-actin is being generated at the boundaries and flowing inward.

2) The point of the talin experiments eluded me. Is the point simply that not everything is going with the flow? If so, having double labels would be more convincing.

3) The Dsh labeling was far below the standard set by the authors in the other images.

*Reviewer #2*:

This manuscript examines the process of notochord formation in *Ciona*, focusing on the positioning of the actomyosin contractile ring. The imaging is fantastic and the long-term drug treatments are rather illuminating and show very non-trivial dynamics. This reviewer was quite satisfied with the quality of experimental data, but the model, while being perfectly sensible, did not seem to produce any non-trivial predictions enabling a serious test of understanding.

The first claimed prediction of the model (in the subsection “A simple biophysical model of the actomyosin cortex for the formation and maintenance of equatorial actomyosin ring”) is that “contractility increases smoothly as elongation and ring migration proceeds”. Since contractility “chi” is a parameter of the model, rather than a dynamical variable, this looks to me like an assumption rather than a prediction.

The second “prediction” of the model that “the velocity gradient of actin filaments towards the center should be proportional to the local contractility, indicated by local actin concentration.” This is such a trivial consequence of force balance (obtained by integration of [Disp-formula equ2]), that it is difficult to accept it as a “prediction”. Instead I would regard this fact, shown in Figure 3, as the basic premise underlying the formulation of the model. A more convincing logical progression would be a) the data in Figure 3 supports the model, b) now calculate non-trivial consequences of the model based on these assumptions that explain a difficult to understand observation.

The fourth paragraph of subsection “A simple biophysical model of the actomyosin cortex for the formation of equatorial actomyosin ring” comes closer to stating model predictions. Figure 3 shows the simulated time evolution of actin intensity. What controls the width of the final steady state (in red)? If a certain set of parameters allows the final predicted profile to fit the observed actin intensity distribution (as shown in Figure 3), does the same set of parameters predict the observed time-evolution of actin intensity before a steady state is achieved? Or, stated in a different way, do the blue, green, and cyan curves match up with the data? Same question can be asked of the actin velocity profiles. Does the predicted time evolution match up with the data?

In the same subsection, the authors state: “Moreover, a rather intriguing feature of our model is that it predicts the size of the actin ring should scale with the length of the cell. This is because ring formation is a spontaneous, self-organizing phenomenon that depends on boundary conditions. We therefore measured experimentally actin ring width during cell elongation, and found it indeed increases linearly with the length of the cell (Figure 3).” Yet, Figure 3 quite clearly shows that the dependence is not linear. Furthermore, the model prediction itself is nonlinear, and this seems to contradict what is stated in the manuscript.

Again, in the same subsection, the authors state: “to probe how PCP can participate in the repositioning of actin rings, we incorporated polarity in the model in the simplest way possible”. The alteration or addition to the model is that PCP alters actin polymerization locally. Simulations of the model with this additional assumption show that the equatorial ring can transition smoothly to an anterior position. The observation that the transition is smooth is not a strong validation of the model; it would be consistent with a whole class of models. A more stringent test of the model is needed. Does the predicted time series of the transition match the data?

In conclusion, I do not see that the model has been tested rigorously. Support for model assumptions is confused with predictions. Centering by contractility seems to be the key insight of the model. It is not too surprising that inclusion of PCP-induced polarity would generate an asymmetry. I believe that the authors could strengthen model “validation” by comparing observed dynamics with model predictions.

*Reviewer #3*:

This study proposes a very interesting and rather novel mechanism for the positioning of a contractile actomyosin ring in *Ciona* notochord cells. However, besides this intriguing finding and the dissection of the individual contributions of contractility and PCP, there are some major issues (also concerning the model) as mentioned next.

1) In Figure 3, the authors measured the contact angle between basal and lateral surfaces and incorporated an increase in contractility (by 2.5) into the model. It is not clear how the contractility has been calculated from the angle measurements. Furthermore, by measuring the contact angle between cells (which is not the same as angle θ between basal and lateral domain as indicated here) the relative cortical tensions can be obtained. Therefore, probably the correct term that should be used here is 'cortical tension' along the basal and lateral domains, which is different from 'contractility'. 'Contractility' arises from active tension generated by actomyosin and 'cortical tension' is the total mechanical tension within the cortex. The authors should clearly state what they measured and how they used the obtained values for the model fitting. The statement of 'Basal contractility increases = angle θ opens' does not make sense, since θ, as indicated in the schematic diagram 3B, is getting smaller in the case of higher 'contractility'. In addition to measuring the contact angles, cortical laser ablation could be used to measure the relative changes of cortical tension throughout time and elongation process.

2) To account for the contribution of PCP, a preferential actin polymerization at the anterior pole was incorporated into the model by changing one boundary condition at the anterior pole from no flux to flux>0 (*J*(0)=0 to *J*(0)=*F*). However, the authors also state that they observe large, non-zero values for the actin filament velocity at both edges of the cell via PIV. Please clarify why it is legitimate to use only anterior flux for model fitting then.

3) Besides affecting actin polymerization there is evidence, that the Wnt/PCP pathway also modulates myosin contractility via Rho GTPases. The authors should make clear why they only proposed polymerization changes in the model and provide more supporting data to show that the influence of PCP is happening indeed via actin polymerization only. The authors further state that Dsh relocated from basal to lateral domains with preference to the anterior side and 'this relocation occurred roughly at the same time as the appearance of the anterior ring (19-20h)'. Figures 1 and 6 show examples of ring appearance around 21hpf (Figure 1) in wildtype case & Dsh lateral localization only at 22 hpf (Figure 6). The authors should provide a better temporal analysis of the proposed components involved. When does the ring appear on average (statistics would be useful) and when is Dsh and Pk localized there? Without providing these data the statement about the actomyosin positioning via “*Prickle/Dishevelled*-mediated anterior bias” cannot be made.

4) In the model, blebbistatin treatment is considered as downregulation of contractility in notochord cells, which leads to an anterior shift of the ring by keeping the inner-ring dynamics persistent (Figure 2). Since blebbistatin not only interferes with contractility but also with other properties such as viscosity, it is very likely that the ring dynamics are changed, which can also be seen by FRAP experiments (half time more >2.5 fold). Therefore, blebbistatin treatment should not only be considered as downregulation of contractility. In addition, blebbistatin treatment interferes with contractility in the whole organism and will change various parameters also in surrounding tissues. It is essential to show the same outcome of actomyosin repositioning with more localized methods, e.g. by using photoinducible aryl azide derivative of blebbistatin to stably inhibit myosin within notochord cells.

5) The model for *aimless* mutant with low polarity and high contractility predicts the ring to form directly in the center of the cell. No data on a time series is shown to support this statement. The model prediction of *aimless* embryos says that the anterior Flux Ja is considerably lowered. Can this be checked experimentally with PIV? It would be good to show what happens to Dsh localization in *aimless* mutants to support the proposed “*Prickle/Dishevelled*-mediated anterior bias”.

[Editors’ note: what now follows is the decision letter after the authors submitted for further consideration.]

Thank you for submitting your work entitled “Assembly and positioning of actomyosin rings by contractility and PCP” for peer review at *eLife*. Your submission has been favourably evaluated by Vivek Malhotra (Senior Editor), Mohan Balasubramanian (Reviewing Editor) and two reviewers.

The reviewers have discussed the reviews with one another, and the Reviewing Editor has drafted this decision to help you prepare a revised submission.

The revised manuscript is improved in quality, however some issues remain unaddressed. The reviewers’ comments are attached for your reference. It appears that most of the issues can be addressed by rewriting and providing explanations. As a result, we have not prioritized the points raised by the referees as to ones that need addressing vs. those that do not. We hope you can address all the points and improve the text accordingly.

Reviewer #1:

The manuscript has been revised along the lines suggested by the different referees. This has considerably improved the study. However, there are still several points of criticism that need to be addressed before the manuscript is suitable for publication.

1) The initial value for contractility was a 2.5-fold increase, which is reflecting the change in contractility during the whole process of ring movement and cell elongation. Why did the authors decide to use the lower 1.5-fold change in contractility (during ring movement only) in the revised version of the manuscript? Please comment on the rationale behind changing this value, and would a 2.5-fold contractility increase instead change the model predictions significantly? (In the figure description of Figure 7, the authors still mention a 2.5-fold linear increase in contractility; is this a typo?)

2) Why did the turnover time measured from FRAP experiments change significantly from 50sec to 90sec in the revised manuscript after re-analyzing data? Have new experimental measurements been done?

3) In Figure 5, when calculating the basal tension from the contact angles, what justifies to neglect adhesion tension without measuring/calculating it for this specific tissue?

There are no measurements of lateral tension, so why is it valid to take lateral tension as constant? Especially in Figure 1, the length of the lateral domain seems to decrease during cell elongation. (Change “basal contractility” to “basal tension” and “Contractility” in Figure 5 to “relative tensions”.)

4) When changing assumptions for boundary tension to S=0.95C (5.2.2) the average filament velocity is not in good agreement with the model prediction anymore. Since in this case the model prediction changes quantitatively, the assumption of S=C might be too much of a simplification?

5) Is polarity still intact in blebbistatin treated embryos? Since the paper by [42] shows the incorrect localization of PCP components (Prickle) upon blebbistatin treatment. In case polarity is disrupted, would blebbistatin treatment then not reflect decrease in contractility in addition to loss in polarity, whereas in the model decrease in contractility will not at the same time lead to a change in polarity?

6) The question of why the authors focused on actin polymerization and did not consider modulations of myosin contractility was not addressed (point 3 in previous comments).

7) Suggestions for *aimless* mutant (point 5 in previous comments) have not been sufficiently addressed.

Reviewer #2:

The manuscript describes an experimental and mathematical modeling study of the assembly and migration of an actomyosin ring in *Ciona* notochord cells, and how these processes depend on planar cell polarity. Confocal fluorescence microscopy showed that after convergent extension basal cortical actin rings assembled close to the anterior pole of the cell. The rings then migrated to the equator. Using the myosin-II ATPase inhibitor blebbistatin and the deletion mutant (“*aimless”*) of a core planar cell polarity component named *prickle*, the authors studied the dependence of ring positioning and directional migration on myosin-II activity and on planar cell polarity. A mathematical model of the basal cortex, based on an active gel model, reproduced experimental ring formation and migration behavior.

Images of actin in *Ciona* notochord cells marked by Lifeact-mEGFP and mCherry-UtrCH showed an actin ring initiating near the anterior pole of the coin-shaped cell ∼17 hours post fertilization, which migrated to the equator as the cell elongated after convergent extension. The lateral distance from ring to anterior (posterior) pole increased (remained constant) as the ring migrated (Figure 1). In cells treated with blebbistatin before ring migration, the ring failed to migrate and remained close to the anterior pole (Figure 1). Blebbistatin treatment after the ring had migrated to the equator caused the ring to return to the anterior pole and eventually disappear. Prolonged blebbistatin treatment produced a second and a third ring (Figure 2).

Experiments with notochord cells of *Ciona* and the ascidian *Halocynthia roretzi*, whose nuclei are positioned on the posterior side and in the center, respectively, both showed ring migration towards the anterior pole after blebbistatin treatment. This suggested to the authors ring motion was likely unaffected by nucleus position.

For *Ciona*, in the *aimless* mutant (which lacks normal polarity) ring motion was random and uncorrelated with nucleus position after treatment with blebbistatin. This contrasts with the wild-type behavior where the ring returned to the anterior pole after blebbistatin treatment, suggesting to the authors that anterior ring migration depends on planar cell polarity (Figure 4).

A 1D mathematical model attributed the rate of change of actomyosin density in the cortex to turnover, gradient of actomyosin flux and a diffusion-like term representing nonlocal turnover ([Disp-formula equ1]). The cortical stress sums contributions from the velocity gradient and actomyosin contractility ([Disp-formula equ3]). Zero external forces on the cortex are assumed, implying constant cortical stress ([Disp-formula equ2]). With low actomyosin contractility and zero flux at the poles (mimicking the cortex before convergent extension) a constant actomyosin density was predicted as seen experimentally. With increased contractility, nonzero anterior flux and zero posterior flux, a “ring” formed (a high density band) near the anterior pole, which migrated slightly beyond the equator. On decreasing the contractility, the ring migrated back to the anterior pole. Both effects agree with experiment, provided one assumes that lowering the contractility correctly represents the effect of blebbistatin (Figures 5 and 6). When the model was run with (1) zero flux at both poles (no polarity), or (2) flux at the posterior pole only, the ring located to (1) the equator or (2) close to the posterior pole (Figure 7).

Based on these results, the authors proposed a self-reinforcing ring formation mechanism driven by actomyosin contractility, and a ring-positioning mechanism depending on unbalanced anterior and posterior boundary actin fluxes (Figures 5 and 7).

This is an interesting study of an actomyosin phenomenon that involves a ring in some respects reminiscent of the cytokinetic contractile ring. The development and migration of the ring is worthy of study, and this paper makes a serious attempt to establish the mechanism involved. The lessons are likely to be generally informative for actomyosin contractility phenomena in the cell cortex, and may lead to insights into cytokinesis.

Some of my major concerns are that insufficient evidence was presented for critical assumptions of the model, and presentation of both experiment and modeling is poor and difficult for a reader to follow. Specific points follow:

1) The paper lacks necessary background and motivation (this would mainly be in the Introduction, but could permeate the paper). The biological significance of the actomyosin phenomenon under study is not explained. The reader does not discover how the actomyosin ring under study is related to elongation. What is the mechanism? A reference to [53] is inadequate. Why should we be concerned about the migration of this ring?

2) In the figures, there is no sign of actin away from the ring and the poles. There is presumably an extended actomyosin cortex everywhere (assumed by the model) yet we don't see it in images. Why is it invisible?

3) In the model, a 3 dimensional stress σ is evolved in space and time. And yet variations of σ in the direction normal to the cell surface are not addressed. Presumably, there is an unstated assumption that variations of σ across the cortex are small and can be neglected? This (or whatever the assumption is) should be clearly stated and it should be explained how this leads to the model equations.

4) The model assumed a nonzero actomyosin flux at the anterior side. This boundary condition was the feature of the model that caused migration and positioning of the ring near the equator. The authors claim this assumption is “supported by the well-studied link between PCP and actin polymerization, notably via Disheveled, which regulates actin assembly”, citing the study of Kida et al. (in the subsection “A simple biophysical model of the actomyosin cortex for the formation and maintenance of equatorial actomyosin ring”). Given the critical role of this assumption, it requires a much more explicit justification that can be properly judged by a reviewer and readers of the journal. What does actin polymerization have to do with it, and how does actin polymerization lead to actomyosin flux, etc.? A detailed, self-contained explanation is needed. A reader should be able to discover why the flux boundary condition is the uniquely correct one, without studying Kida et al.

5) The assumption that basal stress σ¯ increases ∼ 1.5 fold during ring migration is apparently based on a force balance using the measured angle θ between the lateral and basal sides (in the subsection “A simple biophysical model of the actomyosin cortex for the formation and maintenance of equatorial actomyosin ring”, Appendix 4). No data is presented to justify this, neither is any actual force balance equation presented. The validity of the claim is impossible to judge as a result.

6) Related to (2), is it reasonable to assume that γ is constant in time? What is the justification? This was used to deduce the value of the stress σ¯.

7) The model presentation could be improved. The entire model (i.e. equations) plus boundary conditions (essential to all of the results) should be clearly laid out in sequence. At present, the initial conditions and boundary conditions are often not stated or buried in text. Examples:

a) In the fourth paragraph of subsection “A simple biophysical model of the actomyosin cortex for the formation and maintenance of equatorial actomyosin ring”, the initial condition is not stated although the boundary conditions are in the previous paragraph.

b) In the fifth paragraph of the aforementioned subsection, neither initial nor boundary conditions are stated. The key assumptions of their model (third paragraph) were presented after the model was applied to the homogeneous cortex case (second paragraph). Further, it is unclear if the length of the ring is fixed as an input of the model or evolves dynamically.

---

## [Author Response]

[Editors’ note: the author responses to the first round of peer review follow.]

An earlier version of this manuscript was reviewed at *eLife* and was rejected after external review. The decision letter expressed interest in our study but felt our paper suffered from weakness in the modeling part. We agree with the editorial opinion completely that our study should be seen as a work of experiments and modeling working in concert to reveal a novel mechanism in cytoskeleton biology. We also appreciate the constructive and detailed comments and criticisms from the reviewers, which have led to a major revision of our work.

In the new manuscript, we have followed the suggestions of the referees, and divided the physical model part into model assumptions and model predictions. We have expanded the modeling part of the manuscript to two theory figures, Figures 5 and 6, to more clearly articulate the logic progression in our modeling, and to include more analyses, some of which were inspired by referees’ criticisms. Specifically, in this new manuscript we added a much more precise description of the dynamical implications of the model in the main text. Figure 6 compares dynamic data and model at the anterior and posterior positions of the ring, and explores the detailed spatial profiles in intensity, both in normal and blebbistatin-induced migration. We also added another new dynamical check of the model, to show that the decrease in actin filament density subsequent to blebbistatin treatment matches very well quantitatively our predictions. The new analyses and thorough re-work of our model have strengthened our belief that the current work now makes non-trivial predictions, and hypotheses generated have been tested for the modeling to be useful.

We hope you find that our new paper has made substantial improvement, and that the conceptual novelty of our theoretical work, inspired and verified by our novel experimental findings, is now more clearly accentuated.

Reviewer #1:

*1) Based both on the movies and on the fact that many F-actin nucleators concentrate at cell-cell junctions, I was puzzled by the assumption that “actin fluxes vanish” at cell-cell junctions (Results) underlying the model. Am I just misunderstanding what is meant by flux in this context? Perhaps some clarification is in order, as it seems manifestly obvious that F-actin is being generated at the boundaries and flowing inward*.

Prompted by the question, as well as the one from Reviewer 3, we now explain more precisely the meaning of flux in the paper: it consists of both the actual actin flows (\rho v), as well as a non-local term due to actomyosin turnover and diffusion (D d\rho/dx). Therefore, even if we assume that the total flux F vanishes at the posterior boundary, there can still be a flow \rho v of actomyosin, which is counterbalanced by preferential polymerization at the boundary (which is the case in Figure 5 in the main text). We chose the assumption of total flux to vanish because it is the simplest one as a first approximation, and one does not need a non-zero total flux to explain the data. Though we consider the case where the total anterior flux vanishes to explain why contractility drives the formation of a central ring, our final model accounts for a non-zero anterior flux representing the cell polarization. Nevertheless, in order to show more clearly that our mechanism and results hold qualitatively in the presence of a non-zero posterior flux, we show the robustness of our numerical simulations in that case.

*2) The point of the talin experiments eluded me. Is the point simply that not everything is going with the flow? If so, having double labels would be more convincing*.

Indeed, we showed talin data in the manuscript simply to state that not everything is going with the flow. We have live-imaging data showing that talin concentrates at the cell equator slightly after the ring has been established centrally (data not shown), suggesting that talin actually responds to cortical repositioning rather than driving it. Moreover, this data also indicates that there is not a global effect in the embryo after the drug treatment that would shift everything around.

The figure does show a double labelling experiment, where we co-expressed mCherry-talin and Lifeact-emGFP, and present two channels in separate panels.

Nevertheless, we agree that the talin data presented in the main text does not connect well with other parts. To make our story concise and coherent, we now put it into the supplementary material (as Figure 2—figure supplement 2) and shortened the description in this part, while making above point more clear.

*3) The Dsh labeling was far below the standard set by the authors in the other images*.

We admit the quality of these pictures is not at a high level. This is because the images were made using fluorescence microscope that was available at the time. The construct used in this experiment is no longer available. Newer fluorescent protein-fusion constructs have proved to be highly toxic to *Ciona* notochord cells, and either wild type or mutant *dsh* renders notochord abnormal and embryo dead.

We went through all the data and found that these are the best we can offer. We think these images, together with the images in previous publication (24), and recently published data from Smith lab (42) are adequate in support of our claim in the text (we have cited these papers).

Reviewer #2:

*The first claimed prediction of the model (in the subsection “A simple biophysical model of the actomyosin cortex for the formation and maintenance of equatorial actomyosin ring”) is that “contractility increases smoothly as elongation and ring migration proceeds”. Since contractility “chi” is a parameter of the model, rather than a dynamical variable, this looks to me like an assumption rather than a prediction*.

*The second “prediction” of the model that “the velocity gradient of actin filaments towards the center should be proportional to the local contractility, indicated by local actin concentration.” This is such a trivial consequence of force balance (obtained by integration of*
[Disp-formula equ2]*), that it is difficult to accept it as a “prediction”. Instead I would regard this fact, shown in*
Figure 3*, as the basic premise underlying the formulation of the model. A more convincing logical progression would be a) the data in*
Figure 3
*supports the model, b) now calculate non-trivial consequences of the model based on these assumptions that explain a difficult to understand observation*.

We agree with the referee that these experiments are better characterized as validation of model assumptions. We re-organized the text in order to distinguish validations and predictions.

In the first case, we would like to argue that we do in fact measure the rescaled basal tension, related to “chi”, by our angle analysis, following the ideas of Maitre et al. (Science, 2012), so we do see dynamical variation of “chi”.

In the second case, it is worth mentioning that even though the proportionality is a direct consequence of force balance and a visco-contractile rheology, it hadn't been verified before, to the best of our knowledge.

*The fourth paragraph of subsection “A simple biophysical model of the actomyosin cortex for the formation of equatorial actomyosin ring” comes closer to stating model predictions.*
Figure 3
*shows the simulated time evolution of actin intensity. What controls the width of the final steady state (in red)? If a certain set of parameters allows the final predicted profile to fit the observed actin intensity distribution (as shown in*
Figure 3*), does the same set of parameters predict the observed time-evolution of actin intensity before a steady state is achieved? Or, stated in a different way, do the blue, green, and cyan curves match up with the data? Same question can be asked of the actin velocity profiles. Does the predicted time evolution match up with the data*?

We thank the referee for this comment, which allowed us to substantially improve our manuscript.

First, to clarify our model, we now split the theory figures into two: one addressing model assumptions (Figure 5) and a second addressing model predictions (Figure 6).

The width of the ring is dictated by: 1) the cell length and 2) the length scale created by diffusion D and turnover τ, as we show in Appendix 5–figure 3, where we vary D to show the evolution in ring width.

In order to address the second part of the question, we measured dynamically the positions of the anterior and posterior boundaries of the ring and compared to the model, both in the normal migration from anterior to middle, and the reverse migration following blebbistatin treatment (Figure 6), and we found a very good dynamical agreement as well. In order to provide a further test of the model, we compared the theoretical predictions for filament velocity during blebbistatin treatment, and saw an excellent quantitative agreement, although all parameters were already fixed from the profiles of overall ring migration. Finally, we measured, as the referee suggested, the intensity profiles dynamically during both ring migrations, and compared them to the model prediction; again we obtained a good agreement for the same set of parameters. The unique set of parameters taken to fit the data is given and explained in details in the appendices. We have now included a substantial amount of dynamical verification, all for a fixed parameter set. We thank the referee for advising us towards these new analyses.

*In the same subsection, the authors state: “Moreover, a rather intriguing feature of our model is that it predicts the size of the actin ring should scale with the length of the cell. This is because ring formation is a spontaneous, self-organizing phenomenon that depends on boundary conditions. We therefore measured experimentally actin ring width during cell elongation, and found it indeed increases linearly with the length of the cell (*Figure 3*).” Yet,*
Figure 3
*quite clearly shows that the dependence is not linear. Furthermore, the model prediction itself is nonlinear, and this seems to contradict what is stated in the manuscript*.

We apologize for the slight confusion introduced by the term “scaling”. For small and intermediate cell sizes, the ring width indeed scales with cell length, and the dependence is linear. For large cell size, the width increases less rapidly than the cell size indeed.

Nevertheless, this actually matches very well the biological data itself, which displays a plateau for large cell length. We therefore clarified this in the text.

*Again, in the same subsection, the authors state: “to probe how PCP can participate in the repositioning of actin rings, we incorporated polarity in the model in the simplest way possible”. The alteration or addition to the model is that PCP alters actin polymerization locally. Simulations of the model with this additional assumption show that the equatorial ring can transition smoothly to an anterior position. The observation that the transition is smooth is not a strong validation of the model; it would be consistent with a whole class of models*. *A more stringent test of the model is needed. Does the predicted time series of the transition match the data*?

As described above, we performed our analysis on the time series to show how the shape of the actin density profiles reproduce the experimental data, even though there is some biological variations on actin fluorescence levels. We agree that the smoothness of the transition itself is not the proof. Rather, we take a very simple model amenable to detailed mathematical analysis, and show that it can reproduce our data quite accurately. We clarified this in the revised manuscript.

*In conclusion, I do not see that the model has been tested rigorously. Support for model assumptions is confused with predictions. Centering by contractility seems to be the key insight of the model. It is not too surprising that inclusion of PCP-induced polarity would generate an asymmetry. I believe that the authors could strengthen model “validation” by comparing observed dynamics with model predictions*.

We would like to argue that a strength of our model is that we use one of the most simple model imaginable for actomyosin gels, yet this still reproduces very faithfully virtually all the features of our data. This therefore demonstrates the broad applicability of our approach, which should be relevant to the many other biological event involving actomyosin positioning and PCP.

Following the advice of the referee, we improved the manuscript further by comparing the dynamical evolution of the actin intensity between theory and experiments, both in the case of normal and blebbistatin-induced migration (Figure 5 and Figure 6). We've also shown new predictions such as actin velocity evolution, which fit very well quantitatively with our model (Figure 5).

Reviewer #3:

*This study proposes a very interesting and rather novel mechanism for the positioning of a contractile actomyosin ring in* Ciona *notochord cells. However, besides this intriguing finding and the dissection of the individual contributions of contractility and PCP, there are quite some major issues (also concerning the model) as mentioned next*.

*1) In*
Figure 3*, the authors measured the contact angle between basal and lateral surfaces and incorporated an increase in contractility (by 2.5) into the model. It is not clear how the contractility has been calculated from the angle measurements. Furthermore, by measuring the contact angle between cells (which is not the same as angle θ between basal and lateral domain as indicated here) the relative cortical tensions can be obtained. Therefore, probably the correct term that should be used here is 'cortical tension' along the basal and lateral domains, which is different from 'contractility'. 'Contractility' arises from active tension generated by actomyosin and 'cortical tension' is the total mechanical tension within the cortex. The authors should clearly state what they measured and how they used the obtained values for the model fitting. The statement of 'Basal contractility increases = angle θ opens' does not make sense, since θ, as indicated in the schematic diagram 3B, is getting smaller in the case of higher 'contractility'. In addition to measuring the contact angles, cortical laser ablation could be used to measure the relative changes of cortical tension throughout time and elongation process*.

The referee is perfectly right in pointing out that the figure should read “=angle θ closes”, we apologize for this typo due a change in angel definition.

We also clarified how we extract contractility, and the rationale behind it. The absence of a friction term in the force balance equation implies that the cortical tension in the z direction is homogeneous inside the cortex. From the measurement of contact angles, we obtain the increase of such tension as a function of time. A simplifying hypothesis of the model is that this tension (which being homogeneous is also the average tension in the z direction) is proportional to the average contractility (but not the local contractility which as the reviewer rightly underlines is very different, as the viscous part of the stress also enters there).

Within our model, this is a natural hypothesis to make since, in the absence of actin flux at the boundary and when contractility is weak as in stage prior to the ring migration, we expect to have a homogeneous concentration of actin inside the cell which can only be fulfilled if S = C. It is however true that we may have an average tension S slightly smaller than C. We examined the effect of such deviation in the appendices of the revised manuscript, redoing all the numerical integration, and show that our mechanism remains qualitatively unchanged.

Cortical laser ablation is indeed a good tool to measure the cortical tension. We actually tested this technique on *Ciona* notochord several times. However, it does not work well, because the notochord locates at the centre of the tail, surrounded by muscle and epithelial tissues. Increase of laser power will cause damage to the whole tissue/cell structure, making any analysis virtually impossible.

*2) To account for the contribution of PCP, a preferential actin polymerization at the anterior pole was incorporated into the model by changing one boundary condition at the anterior pole from no flux to flux>0 (*J*(0)=0 to* J*(0)=*F*). However, the authors also state that they observe large, non-zero values for the actin filament velocity at both edges of the cell via PIV. Please clarify why it is legitimate to use only anterior flux for model fitting then*.

In this model, we make a distinction between the actin flux and velocity. Indeed, the flux also contains a diffusive part, which effectively accounts for the actin production in the layer as we have now clarified in the main text, and explained in greater details in the appendices.

As a consequence, while the total flux may vanish at the boundary, (*J*(*L*) = 0), indicating that there is no local overall production at this side, the velocity is not necessarily zero, because of actin polymerization which can counterbalance the velocity of actin filaments going in.

Nevertheless, the referee is correct that the total flux could also be non-zero. Therefore, we also now show the robustness of our results in the presence of a small flux at the posterior side. We show that again our mechanism is qualitatively unchanged, as only the difference between anterior and posterior boundary fluxes matter. We clarified the text by adding a description as following:

“Interestingly, we predicted, and observed in the PIV, a rather large, non-zero value for the actin filament velocity at the posterior edge of the cell, which is compensated in the model by actin polymerization on the side, since the total flux is zero. Moreover, we note a remaining, albeit small bias in ring position towards the anterior side, as in the data, showing that an anterior-imbalance remains at the later stages, both in simulations and in the data.”

*3) Besides affecting actin polymerization there is evidence, that the Wnt/PCP pathway also modulates myosin contractility via Rho GTPases. The authors should make clear why they only proposed polymerization changes in the model and provide more supporting data to show that the influence of PCP is happening indeed via actin polymerization only. The authors further state that Dsh relocated from basal to lateral domains with preference to the anterior side and 'this relocation occurred roughly at the same time as the appearance of the anterior ring (19-20h)'.*
Figures 1 and 6
*show examples of ring appearance around 21hpf (*Figure 1*) in wildtype case & Dsh lateral localization only at 22 hpf (*Figure 6*). The authors should provide a better temporal analysis of the proposed components involved. When does the ring appear on average (statistics would be useful) and when is Dsh and Pk localized there? Without providing these data the statement about the actomyosin positioning via “*Prickle/Dishevelled*-mediated anterior bias” cannot be made*.

We appreciate the reviewer's criticism. We have now made it clear in the text that two references, [24] and [42], show the time course of Dsh and Pk. The data from these and our current manuscript provide a very detailed time course on the localization of these proteins. We feel these data together support our claim that the anterior relocation of dsh to the anterior edge, which begins to show an anterior bias already at 20.8 h (see Figure 2 middle picture), occurs very early during ring formation.

*4) In the model, blebbistatin treatment is considered as downregulation of contractility in notochord cells, which leads to an anterior shift of the ring by keeping the inner-ring dynamics persistent (*Figure 2*). Since blebbistatin not only interferes with contractility but also with other properties such as viscosity, it is very likely that the ring dynamics are changed, which can also be seen by FRAP experiments (half time more >2.5 fold). Therefore, blebbistatin treatment should not only be considered as downregulation of contractility*.

The reviewer is correct in stating that blebbistatin surely influences more than contractility. Nevertheless, it is widely accepted that we now explicitly state in the manuscript's Discussion the following:

“Indeed, our model is the simplest one can write for active gels, and it assumes for instance that all the rheological coefficients are constant, while in principle some, for example viscosity, could depend on the state of the gel, and on blebbistatin concentration in the drug treatment (56). Nevertheless, the fact that we can capture well the experimental data with such a simple model suggests that such coupling effects are probably of secondary importance for the observed phenomenon.”

One should note, as we did in our revised manuscript, that a 2.5 fold increase in FRAP half-time does not mean a 2.5 increase in turnover time, as blebbistatin decreases the speed of turnover flows, and therefore one would expect, for constant turnover an increased time in ring recovery.

Therefore, to address the reviewer's remark, we re-analyzed quantitatively our FRAP data. Interestingly, we find that although the control data is well-fitted by a single exponential, the blebbistatin recovery is much better fitted by two-exponential (Figure 2), where the fast timescale is very similar to control, and the second much longer. A simple calculation shows that given the measured velocity of flow, one expects quantitatively this behavior. This has been added to the new manuscript as following:

“Indeed, in control cells, a rough estimate of the flow induced time needed to close a bleached segment of 3µm, given a mean velocity of 30 nm/s, is 100s, which cannot be distinguished from the turnover time. Therefore, we only see a single-exponential recovery. […] There are then two time scales for recovery, one linked to turnover, unaffected by blebbistatin, and one linked to flows, which are dramatically compromised by blebbistatin.”

*In addition, blebbistatin treatment interferes with contractility in the whole organism and will change various parameters also in surrounding tissues. It is essential to show the same outcome of actomyosin repositioning with more localized methods, e.g. by using photoinducible aryl azide derivative of blebbistatin to stably inhibit myosin within notochord cells*.

We appreciate the reviewer's concern that the whole organism, including the tissues directly in contact with notochord cells, is also treated with the drug. We have indeed sought out methods to disrupt contractility locally and specifically in the notochord cells. As stated above, we have discovered that cortical laser ablation is technically impossible in *Ciona* notochord cells.

We have also looked into azidoblebbistatin and found that its applicability in live embryos is limited, and the drug is hard to obtain. Furthermore, we have found out other species of commercially available blebbistatin behaved erratically in cells and in embryos. For example, nitro-blebbistatin formed numerous, large, and highly fluorescent precipitants in the cytoplasm, probably due to certain reactions to the salinity or certain content of seawater.

Interestingly, in a separate line of experiments conducted aiming to search for an external cause of the ring shifting, i.e. the surrounding tissues are driving the ring to migrate to the anterior end of the notochord cells, we found the relative movement of notochord cells and its neighbouring tissues was not disrupted by blebbistatin. For example, the extension and posterior migration of muscle proceeded normally. Under the microscope the relatively sliding of muscle cells over notochord was undeterred. Most importantly, there was no correlation between the movement of the muscle and the anterior migration of the ring (unpublished observations). Based on these observations, we believe the ring behaves quite autonomously within the notochord cells. The relatively autonomy of notochord in the morphogenesis of *Ciona* embryonic development is supported by both genetics and embryology studies, and the higher sensitivity of notochord cells may be underscored by the highly abundant expression of myosin- and actin-regulatory proteins in notochord compared with other tissues.

*5) The model for* aimless *mutant with low polarity and high contractility predicts the ring to form directly in the center of the cell. No data on a time series is shown to support this statement. The model prediction of* aimless *embryos says that the anterior Flux Ja is considerably lowered. Can this be checked experimentally with PIV? It would be good to show what happens to Dsh localization in* aimless *mutants to support the proposed “*Prickle/Dishevelled*-mediated anterior bias”*.

We now added a much more precise description of the dynamical implications of the model in the main text. We have added a figure (Figure 6), where we compare dynamically data and model on the anterior and posterior positions of the ring, as well as on the detailed spatial profiles in intensity, both in normal and blebbistatin migration. We also added another new dynamical check of the model, to show that the decrease in actin filament density subsequent to blebbistatin treatment matches very well quantitatively our predictions.

The model for *aimless* mutant does not necessarily predict that the ring forms exactly at the center since there may be some imbalance between the small remaining posterior and anterior fluxes that make one side prevail. Nevertheless, the localization is expected theoretically to be much less precise, which is what we see experimentally.

[Editors' note: the author responses to the re-review follow.]

Reviewer #1:

*1) The initial value for contractility was a 2.5-fold increase, which is reflecting the change in contractility during the whole process of ring movement and cell elongation. Why did the authors decide to use the lower 1.5-fold change in contractility (during ring movement only) in the revised version of the manuscript? Please comment on the rationale behind changing this value, and would a 2.5-fold contractility increase instead change the model predictions significantly? (In the figure description of*
Figure 7*, the authors still mention a 2.5-fold linear increase in contractility; is this a typo?*)

We had shown that there was a total 2.5-fold and 1.5-fold increase in contractility during the process of notochord cell elongation and the early process of ring migration, respectively (in the subsection “A simple biophysical model of the actomyosin cortex for the formation and maintenance of equatorial actomyosin ring” and in Figure 5). Nevertheless, in the simulation (Figure 6), we concentrated on the early process of ring migration, which reaches the center before the end of elongation, corresponding to a 1.5-fold increase only, hence the change. We modified Figure 5 and figure legend to indicate clearly 1.5 and 2.5-fold increase on the relative contractility at 2 different stages.

Assuming a 2.5-fold increase would not change qualitatively the story, but assuming that other parameters remain unchanged, would make the migration faster, as well as the ring more peaked.

In Figure 7 legend, this is indeed a typo. It should be 1.5-fold linear increase in contractility.

*2) Why did the turnover time measured from FRAP experiments change significantly from 50sec to 90sec in the revised manuscript after re-analyzing data? Have new experimental measurements been done*?

The first version of the manuscript mentioned 50s as being the half-time of recovery. Nevertheless, we realized that the turnover involved in the model is the exponential recovery time (which is by definition 30% higher than the half time). We also re-analyzed the data as mentioned in the previous response letter to correct for some photo-bleaching and the presence of one recovery time for the control versus two recovery times for the blebbistatin treated cells. We explain this better and give the detailed results from the fits in the legend of Figure 2.

*3) In*
Figure 5*, when calculating the basal tension from the contact angles, what justifies to neglect adhesion tension without measuring/calculating it for this specific tissue*?

*There are no measurements of lateral tension, so why is it valid to take lateral tension as constant? Especially in*
Figure 1*, the length of the lateral domain seems to decrease during cell elongation. (Change “basal contractility” to “basal tension” and “Contractility” in*
Figure 5
*to “relative tensions”*.)

We made the modifications suggested by the referee on the terminology.

Adhesion is very hard to quantify in this system, although pipettes measurements on more easily accessible systems (Maitre et al, Science, 2012; Maitre et al, NCB, 2015) suggests it is negligible for force balance.

We also agree that constant lateral tension is an approximation and that during the ring migration the length (height) of lateral domain indeed decreases, suggesting the increase of lateral domain tension. Unfortunately, right now, it is not possible for us to directly measure the lateral tension in *Ciona* notochord cells. Our measurements provide the ratio of the basal tension over the lateral tension (it should be noted that absolute measurements are very hard to get, as even laser ablation does not measure tension, but tension over viscosity, which can also change). This ratio increases linearly in time and by supposing that the lateral tension is roughly constant, we might underestimate the basal tension increase. We may thus view it as giving a rough approximation of the contractility increase which precise value is adjusted in the theoretical model to fit the experiments. We added a statement to the subsection “A simple biophysical model of the actomyosin cortex for the formation and maintenance of equatorial actomyosin ring”.

We now make this clearer in the text and thank the reviewer for bringing this to our attention.

*4) When changing assumptions for boundary tension to S=0.95C (5.2.2) the average filament velocity is not in good agreement with the model prediction anymore*. *Since in this case the model prediction changes quantitatively, the assumption of S=C might be too much of a simplification*?

We first would like to emphasise that the choice S=C is not arbitrary, as it is the only one leading to a homogeneous actin distribution before elongation (low contractility and polarity) as observed experimentally. We discussed this in more detail in the model presentation.

In the supplementary material, we indeed show that if we perturb the constraint S=C, the actomyosin flow is strongly perturbed which suggests that the tension externally applied by the neighbouring cells strongly regulates the actomyosin flow. Notice however that this does not invalidate the model as, if we fix S≠C, we could still get a good agreement with experiments from adjusting the boundary flux parameter.

The main aim of the model is to expose the simplest mechanical agents driving the migration of the ring. For this, we focus on a single a cell and impose a simple force boundary condition consistent with a homogeneous cortex at low basal contractility and polarity. We thank again the reviewer for suggesting us to investigate if this condition evolves during the elongation process and it is our hope that future works analysing the whole 39 cells assembly will shed more light on this boundary condition.

*5) Is polarity still intact in blebbistatin treated embryos? Since the paper by*
[42]
*shows the incorrect localization of PCP components (Prickle) upon blebbistatin treatment. In case polarity is disrupted, would blebbistatin treatment then not reflect decrease in contractility in addition to loss in polarity*, *whereas in the model decrease in contractility will not at the same time lead to a change in polarity*?

As migration in blebbistatin-treated embryos was always towards same (anterior) side, there is clearly still polarity in the system. Actually we show on Appendix 5–figure 3 that disrupting the polarity in a mild way by adding a small posterior flux does not affect our results on migration at all (in physical terms, an important theoretical result is that the system is locally robust with respect to this type of perturbation). We discussed this again in the presentation of the model.

*6) The question of why the authors focused on actin polymerization and did not consider modulations of myosin contractility was not addressed (point 3 in previous comments)*.

We would like to point out that although our argument relies on actin polymerization, because of its known link to Dsh and PCP, our 1D model is an effective model of both actin and myosin as a gel treated as single specie with the assumption that bipolar filaments performing the contractile power stroke co-localize with actin. Therefore, our assumption that actin is brought preferentially at the anterior side resulting in an effective increased contractility there (because of increased density), which would be very similar qualitatively to considering preferential myosin activity. We made the connection more explicit in our revision and discussed [42] in the Results section and in the Discussion.

*7) Suggestions for* aimless *mutant (point 5 in previous comments) have not been sufficiently addressed*.

The initial suggestion of the reviewer is interesting. However, live-imaging in these conditions is extremely challenging for several reasons. In *pcp* mutants, cell intercalation is disrupted in the notochord. Only few last cells present a normal elongated shape. In addition, until the later stage of embryonic development, we cannot distinguish which cells are PCP mutants (we have no early genetic marker in *Ciona)*.

Reviewer #2:

*1) The paper lacks necessary background and motivation (this would mainly be in the Introduction, but could permeate the paper). The biological significance of the actomyosin phenomenon under study is not explained. The reader does not discover how the actomyosin ring under study is related to elongation*.

We have re-edited the text and added additional explanations in the Introduction. In particular, we have made explicit the connection between ring positioning and cell shape, but also the fact that in cytokinesis, precise ring positioning is crucial to ensure the proper segregation of fate determinants (Clevers et al., Nature Genetics, 2005; Sedzinski et al., Nature, 2011 for instance)

*What is the mechanism? A reference to*
[53]
*is inadequate. Why should we be concerned about the migration of this ring*?

In Sehring et al. (PLOS Biology, 2014), the scenario we present is that the force generated by actomyosin constriction transmits pinches from the basal cortex to the anterior and posterior lateral domains, and therefore elongates the cell because the cytoplasm is incompressible, thus driving notochord elongation.

Crucially, in our previous studies, we found that the position of contractile rings significantly influences notochord cell shape and elongation. For example, in α-actinin mutants, the ring cannot maintain at the equator and the cells hereby fail to elongate and become asymmetric shape (53). This stimulated us to focus on the position of rings, which we clarify in our Introduction.

As the referee correctly points out, it will be intriguing in the future to study quantitatively in more details the interplay between ring positioning, that we elucidate here, and the detailed cell shape.

*2) In the figures, there is no sign of actin away from the ring and the poles. There is presumably an extended actomyosin cortex everywhere (assumed by the model) yet we don't see it in images*. *Why is it invisible*?

This is indeed an illusion: we corrected the background in all the figures showing intensity profiles, and the actin is never zero (typically 1/3 to 1/4 of the peak). We clarified that the cortex is continuous in the introduction to the modelling.

*3) In the model, a 3 dimensional stress σ is evolved in space and time. And yet variations of σ in the direction normal to the cell surface are not addressed. Presumably, there is an unstated assumption that variations of σ across the cortex are small and can be neglected? This (or whatever the assumption is) should be clearly stated and it should be explained how this leads to the model equations*.

We thank the referee for raising this point.

Indeed, we have used the lubrication approximation to integrate over variations in the perpendicular direction to the gel. This is a very common approximation in hydrodynamics, valid if the thickness of the cortex is small compared to the length of the cell. And indeed, typical cortex thickness is 100-200 nanometers, whereas the typical cell length is 10 microns, two orders of magnitude above.

We now made this more explicit in the model presentation.

*4) The model assumed a nonzero actomyosin flux at the anterior side. This boundary condition was the feature of the model that caused migration and positioning of the ring near the equator*.

We would like to point out that the actomyosin flux at the anterior side does not cause the migration per se. It allows the distribution of actin at low contractility to be polarized and not homogeneous. The positioning of the ring to the equator is due to the increased contractility, which shifts the ring to the center. We clarified this in the modelling part.

*The authors claim this assumption is “supported by the well-studied link between PCP and actin polymerization, notably via Disheveled, which regulates actin assembly”, citing the study of Kida et al. (subsection “A simple biophysical model of the actomyosin cortex for the formation and maintenance of equatorial actomyosin ring”). Given the critical role of this assumption, it requires a much more explicit justification that can be properly judged by a reviewer and readers of the journal. What does actin polymerization have to do with it, and how does actin polymerization lead to actomyosin flux, etc.? Detailed, self-contained explanation is needed. A reader should be able to discover why the flux boundary condition is the uniquely correct one, without studying Kida et al*.

We thank the referee for the suggestion and present the link between PCP and actin assembly much more explicitly in the main text. However, we would like to mention that the “positioning of the ring near the equator” mentioned by the referee is not due to boundary condition, but only due to acto-myosin contractility, which we also make more explicit in the new manuscript (and related to comment 7).

*5) The assumption that basal stress*
σ¯
*increases ∼ 1.5 fold during ring migration is apparently based on a force balance using the measured angle θ between the lateral and basal sides (subsection “A simple biophysical model of the actomyosin cortex for the formation and maintenance of equatorial actomyosin ring”,*
Appendix 4*). No data is presented to justify this, neither is any actual force balance equation presented. The validity of the claim is impossible to judge as a result*.

The data to justify this is Figure 5, which displays changes in measured changes θ at different time points. We clarify this point, and write explicitly force balance at the lateral/basal interface in the main text, in order to make the reasoning clear.

*6) Related to (2), is it reasonable to assume that γ is constant in time? What is the justification? This was used to deduce the value of the stress*
σ¯.

As we discussed for the comment of Reviewer 1, this is the simplest hypothesis one can make.

*7) The model presentation could be improved. The entire model (i.e. equations) plus boundary conditions (essential to all of the results) should be clearly laid out in sequence. At present, the initial conditions and boundary conditions are often not stated or buried in text. Examples*:

*a) In the fourth paragraph of subsection “A simple biophysical model of the actomyosin cortex for the formation and maintenance of equatorial actomyosin ring”, the initial condition is not stated although the boundary conditions are in the previous paragraph*.

*b) In the fifth paragraph of the aforementioned subsection, neither initial nor boundary conditions are stated. The key assumptions of their model (third paragraph) were presented after the model was applied to the homogeneous cortex case (second paragraph). Further, it is unclear if the length of the ring is fixed as an input of the model or evolves dynamically*.

We thank the referee for the suggestion, and followed his guidelines to expand the presentation of the theory section.